# Radio: Rate–Distortion Optimization for Large Language Model Compression

Sean I. Young [1] [2]

## Abstract

In recent years, the compression of large language models (LLMs) has emerged as a key problem in facilitating LLM deployment on resource-limited devices, reducing compute costs, and mitigating the environmental footprint due to large-scale AI infrastructure. Here, we establish the foundations of LLM quantization from a rate–distortion theory perspective and propose a quantization technique based on simple rate–distortion optimization. Our technique scales to models containing hundreds of billions of weight parameters and offers users the flexibility to compress models, post-training, to a model size or accuracy specified by the user.

## 1. Introduction

Large Language Models (LLMs) have become a universal framework for solving a wide range of problems in natural language processing, ranging from text summarization and translation to conversational AI. While LLMs have already surpassed traditional methods in many of these tasks, they involve tens to hundreds of billions of weight parameters (!), rendering their deployment onto devices with limited resources challenging—model weights and activations far exceed the available on-chip memory so that weights need to be loaded from the off-chip (global) memory throughout inference, rendering LLM inference memory-bound (Yuan et al., 2024). This greatly hinders the applicability of LLMs particularly in time-sensitive applications and exacerbates the environmental footprint of large-scale AI infrastructure required by LLMs.

One way to reduce the memory requirements of LLMs for inference is by simplifying the model representation post-training. Quantization of the model weights and activation has proven to be particularly apt at compressing models to low bit depths or even arbitrary model sizes (Dettmers et al., 2022; Yao et al., 2022; Frantar et al., 2022; Frantar & Alistarh, 2022; Kim et al., 2024; Shao et al., 2024; Lee et al., 2024; Guan et al., 2024). Using state-of-the-art model quantization techniques, it is possible to compress 10–100

[1] Martinos Center, Harvard Medical School, Boston, MA, USA. [2] Computer Science and Artificial Intelligence Lab (CSAIL), MIT, Cambridge, MA, USA. Correspondence to: Sean I. Young <siyoung@mit.edu>.

*Proceedings of the 42nd International Conference on Machine Learning*, Vancouver, Canada. PMLR 267, 2025. Copyright 2025 by the author(s).

billion-parameter LLMs to 3–4 bits per weight on average and incur only a negligible loss in model accuracy (Chee et al., 2024; Frantar et al., 2022; Lin et al., 2024). This enables LLM inference on a single consumer-grade GPU.

Despite the practical advances made in model quantization methods in recent years, rate–distortion theoretic aspects of LLM quantization are seldom studied in earlier works. By far, the most extensively studied and extended framework for LLM quantization is Optimal Brain Compression and GPTQ of Frantar et al. (2023), itself an adaptation of the classic Optimal Brain Surgery (OBS) algorithm (Hassibi & Storck, 1992). Since OBS operates outside the framework of rate and distortion—which serves as the basis of many related compression problems—a rate–distortion theoretic characterization and exposition of LLM compression can significantly enhance our understanding of the problem at hand and guide the design of LLM compression methods.

In this paper, we tackle the problem of LLM compression using a rate–distortion framework. We begin by analyzing how a model's weights should be quantized to maximize quantized model accuracy at a given average bit depth (bit rate). After this, we propose a stochastic gradient descent-type method to solve this optimization problem exactly and efficiently, post-training—in minutes for billion-parameter models and in a few hours for 10–100-billion-parameter models. Compared with OPTQ and its extensions (Frantar et al., 2022; Frantar & Alistarh, 2022; Huang et al., 2024; Lee et al., 2024; van Baalen et al., 2024), in which weights must be fine-tuned during quantization, our rate–distortion framework more simply determines optimal bit depths and uses integer rounding heuristic for the actual quantization the optimum bit depths have been determined. This renders our framework also suited for quantizing the intermediate activations, which can further reduce the memory footprint of batched inference.

More specifically, our contributions are as follows.

- We formulate a rate–distortion theoretic framework for rate–distortion optimal quantization of LLMs.

- We design a stochastic ascent algorithm for solving the resulting rate–distortion optimization problem.

- We quantize LLMs across model types and sizes and show the rate–distortion behavior of quantized LLMs.

## 2. Previous Work

Earlier work on neural network model quantization can be traced back to Vanhoucke et al. (2011), who demonstrated

that 8-bit integer arithmetic is sufficient for neural network training and inference without incurring a significant loss of model accuracy. In general, quantization-aware training (QAT) (Zhou et al., 2017; Jacob et al., 2018; D. Zhang et al., 2018; Esser et al., 2019; Y. Choi et al., 2017; Wang et al., 2019) integrates the quantization process into training by allowing the model to adapt to the reduced bit precision in weights (Esser et al., 2019; Jacob et al., 2018; D. Zhang et al., 2018; Zhou et al., 2017) and activations (Y. Choi et al., 2017; Wang et al., 2019), determining the optimal bit depth (Wang et al., 2019; D. Zhang et al., 2018) and step size (Esser et al., 2019) using backpropagation to facilitate the flow of gradient through to quantization operators. One shortcoming of QAT methods is that model training needs to be repeated for different quantized model bit depths and accuracy, which can render them less suited for quantizing larger neural network models such as LLMs.

More recent quantization methods for language and vision models aim to facilitate compression of pre-trained models for rapid deployment without additional training (Dong et al., 2019; Chen et al., 2021; Dettmers et al., 2022; Yao et al., 2022; Frantar et al., 2022; Dettmers et al., 2023; Xiao et al., 2023; Lin et al., 2024; Kim et al., 2024; Shao et al., 2024; Lee et al., 2024). These methods quantize weights to 3–4 or 8 bits for integer-arithmetic-only inference (Jacob et al., 2018) by using mixed bit depth quantization (Wang et al., 2019; Chen et al., 2021) or by a separate handling of outlier channels (Zhao et al., 2019) to improve the accuracy of the quantized model. Loss-aware quantization methods (Hou & Kwok, 2018; Nahshan et al., 2020; Qu et al., 2020) seek to minimize the accuracy loss in quantized models by calibrating quantization and biases on a set of calibration examples. Data-free quantization (Nagel et al., 2019; Xu et al., 2020; K. Choi et al., 2021; Qian et al., 2023) attempts to remove the need for real calibration data by matching the distribution of weights instead (Nagel et al., 2019) or using synthetic data in place of real calibration data (K. Choi et al., 2021).

For the compression of LLMs in particular, an extension to the Optimum Brain Surgeon (OBS) algorithm (Hassibi & Stork, 1992) referred to as GPTQ (Frantar et al., 2022) has been proposed for compressing 10–100 billion parameter models. More recent extensions (Dettmers et al., 2023; Lee et al., 2024) to GPTQ incorporate the handling of the more sensitive model weights by re-scaling them or by retaining the original weight values similar to (Lin et al., 2024; Xiao et al., 2023), low-rank decomposition of quantization error matrices (Shao et al., 2024) as well as orthogonal transform of weight matrices prior to their quantization (Ashkboos et al., 2024). While mixed-precision weight quantization is a promising paradigm for handling weights with different sensitivity, current mixed-precision approaches (Wang et al., 2019; Chen et al., 2021; Lee et al., 2024; Dettmers et al., 2023) assign different bit depths from a limited set of bit-depth options (e.g., 4 or 16 bits) or only across different layers. This is due to the combinatorial nature of mixed bit

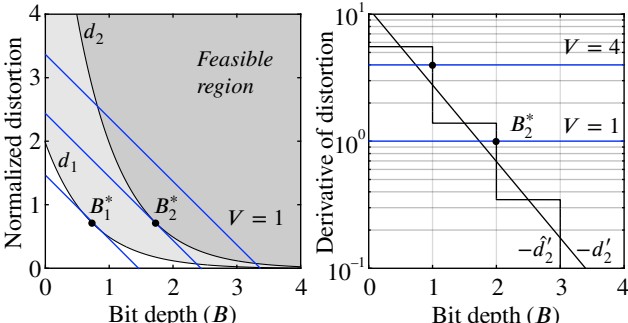

**Figure 1: Optimum bit depths.** Consider two weight matrices such that their distortion functions are given by $d_1$ and $d_2$, where $d_n(B_n) = G_n^2 S_n^2 2^{-2B_n}$. For any value of the dual $V$, optimal bit depths $B_1^*$ and $B_2^*$ are found where the derivative of $d_1$ (resp. $d_2$) is $-V$ (left). These points correspond to the intersections between $V$ and $-d_n' = (2\ln 2) d_n$ (right). Integerized bit depths are on $-\hat{d}_n'$.

depth assignment and limits the attainable quantized model accuracy especially for LLMs with hundreds of billions of parameters. Appealing to a rate–distortion framework can not only overcome the combinatorial nature of determining the optimal bit depths $(0, 1, \dots, 8 \text{ bits})$ at a finer level of granularity (e.g., per channel or per weight group) but also enhance our understanding the LLM compression problem through the lens of rate–distortion theory, which has been fundamental to understanding and solving the quantization problem in image, audio and video domains. Extensions to transform coding and activation quantization are discussed in the sequel.

## 3. Quantization Framework

Here, we use the task of next-token prediction in language modeling as a running example. For our purposes, the end-to-end mapping of input token embeddings to its predicted next-token embeddings by a pretrained language model can be expressed in the most general form as

$$\mathbf{Z} = f(\mathbf{X}) = f(\mathbf{X}, \mathbf{\Theta}_1, \dots, \mathbf{\Theta}_N, \mathbf{b}_1, \dots, \mathbf{b}_N), \quad (1)$$

in which $\mathbf{X} \in \mathbb{R}^{L \times E}$ denotes a sequence of $L$ tokens, each one residing in some $E$-dimensional embedding space, and $\mathbf{Z} \in \mathbb{R}^{L \times E}$ are embeddings of $L$ predicted next tokens. The $m$th block of weight matrices $(\mathbf{\Theta}_{mM+1}, \dots, \mathbf{\Theta}_{(m+1)M})$ and the bias vectors $(\mathbf{b}_{mM+1}, \dots, \mathbf{b}_{(m+1)M})$ parametrize the $m$th transformer block, which refines the embeddings produced by the $(m-1)$th transformer block. LLM frameworks used in language modeling typically also require an embedding matrix $\mathbf{\Theta}_0 \in \mathbb{R}^{E \times V}$ and prediction head $\mathbf{\Theta}_{N+1} \in \mathbb{R}^{V \times E}$ to convert between embeddings and tokens from a vocabulary of some size $V$. In this work, we focus on the compression of transformer block weights as customarily done in most model weight quantization work (Frantar et al., 2022; Lee et al., 2024; Lin et al., 2024).

To get a sense of the number of weight matrices and their sizes in a typical language model, the 13 billion-parameter

**Algorithm 1.** Radio: Rate–Distortion Optimization for LLM Compression

---

1   **Input:** $f(\cdot, \Theta_1, \ldots, \Theta_N)$ (model), $\{X\}$ (calibration set),
2     R (target bit rate), $B_{max} \leftarrow 8$ (max bit depth)
3   **Output:** $B_1, \ldots, B_N$ (bit depths), $S_1, \ldots, S_N$ (weight scales),
4     $\mu_1, \ldots, \mu_N$ (weight means)
5   **Initialize:** $U \leftarrow \text{pca\_basis}(\{X\}) \in \mathbb{R}^{E \times E'}$, $V \leftarrow 10^{-6}$
6     $B_n \leftarrow \infty$, $G_n^2 \leftarrow 0$, $\mu_n \leftarrow \text{mean}(\Theta_n)$, $S_n \leftarrow \text{std}(\Theta_n)$,
7     $\Theta_n^q \leftarrow \Theta_n$, $b_n^q \leftarrow b_n$, $\bar{X}_n \leftarrow 0$ **forall** n
8   **for** iter in $1, \ldots, \text{max\_iter}$ **do**
9      **for** X in minibatch **do**
10        $Z, X_1, \ldots, X_N \leftarrow f(X, \Theta_1^q, \ldots, \Theta_N^q, B_1^q, \ldots, B_N^q)$
11        $\bar{X}_n \leftarrow (1-\alpha)\bar{X}_n + (\alpha/L)\mathbf{1}^\top X_n$ **forall** n
12        $\Gamma_1, \ldots, \Gamma_N \leftarrow \text{autograd}(S^\top Z U, \Theta_1^q, \ldots, \Theta_N^q)$
13        $G_n^2 \leftarrow (1-\alpha)G_n^2 + (\alpha/P_n)\text{trace}(\Gamma_n^\top \Gamma_n)$ **forall** n
14      **for** _ in $1, \ldots, 10$ **do**
15        $B_n \leftarrow \text{clamp}\left(\frac{1}{2}\log_2\left(2\ln 2 \, G_n^2 S_n^2 / V\right), 0, B_{max}\right)$ **forall** n
16        $V \leftarrow V + \beta\left(\text{sum}(P_n B_n) - (\text{sum}(P_n))R\right)$
17      $\Theta_n^q \leftarrow \text{compand\_quantize}(\Theta_n, B_n, S_n, \mu_n)$,
18      $b_n^q \leftarrow b_n + (\Theta_n^q - \Theta_n)\bar{X}_n$ **forall** n

---

model in the Meta OPT family contains $N = 240$ weight matrices in blocks of $M = 6$, with each block comprising $12E^2$ weights in an embedding dimension $E = 5120$. The embedder and prediction head are jointly parameterized by one matrix containing $VE$ weights, where the vocabulary size $V = 50272$. Each transformer block also contains $9E$ bias parameters but due to their relative scarcity, these bias parameters can be communicated losslessly and still have little to no impact on the overall compression performance (Frantar et al., 2022).

Notionally, the elements of $\Theta_n$ are continuously valued so they require quantization for efficient communication and storage. Compared with vector quantization techniques of (Egiazarian et al., 2024; Gong et al., 2015; van Baalen et al., 2024), scalar quantization (Frantar et al., 2022; Lin et al., 2024) simplifies decoding and even enables operations directly on quantization indices, which obviates the need for a separate dequantization process. If mid-rise uniform scalar quantization is used, dequantization of a weight $\theta$ at a bit-depth of $B$ bits and a step size $D$ can be expressed as

$$\theta^q(B, D) = D\left(\text{clip}(\lfloor \theta/D \rfloor, -2^{B-1}, 2^{B-1} - 1) + 2^{-1}\right) \quad (2)$$

for $B = 0, 1, 2, \ldots$, and $\theta^q = \theta$ when $B = \infty$ for notational convenience. The problem of compressing a model $f$ then boils down to determining the optimal bit depth $B$ and the quantization step $D$ for each group of model weights. It is impractical, of course, to determine a separate $(B, D)$ for each weight $\theta$ in the model since the cost of signaling the choice of $(B, D)$ for each one would greatly exceed the bit savings derived from quantization. Typically, a $(B, D)$ pair is used to quantize a small group of weights (e.g., an entire matrix or rows or columns thereof) in which case the cost of signaling $(B, D)$ is borne by a group of quantized weight parameters as a negligible per-weight overhead.

## 3.1. Bit Depth Assignment

Suppose we want to compress $f$ by quantizing each matrix $\Theta_n$ containing $P_n$ elements according to its own bit depth $B_n$ and step size $D_n^*(B_n)$. Generally speaking, weights that are more sensitive to output distortion should be allocated more bits to "balance the scales" while the total number of bits is kept under some model bit budget. We can formalize this notion by expressing the quantization problem at hand as a constrained least-squares problem:

$$\min \ d(\{B_n\}) = \mathbb{E}_X \left\| f(X, \{\Theta_n^q(B_n)\}_{n=1}^N) - f(X) \right\|^2$$
$$\text{s.t.} \ \ r(\{B_n\}) = \sum_{n=1}^N P_n B_n - \left(\sum_{n=1}^N P_n\right)R = 0, \quad (3)$$

in which $R$ is a user-specified average model bit depth (or bit *rate*) and $\Theta_n^q(B_n) = \Theta_n^q(B_n, D_n^*(B_n))$ for brevity. This is a problem similar to optimal resource allocation, where the objective is to maximize some utility (minimizing output distortion in this context) by optimally spending down a given budget (the total number of bits). In this section and next, we provide insights into problem (3) and discuss its optimization and solution; see Algorithm 1.

To apply the machinery of numerical optimization, we can relax the discrete constraint on the bit depths $B_1, \ldots, B_N$ of (3) while solving the problem and round off the solution $B_1^*, \ldots, B_N^*$ to their nearest integers after we have obtained them. Expressing the Lagrangian of (3) as $\mathscr{L}(\{B_n\}, V) = d(\{B_n\}) + Vr(\{B_n\})$, where $V \in \mathbb{R}$ is the dual variable for the equality constraint of (3), we set equal to 0 the partials of $\mathscr{L}$ with respect to $B_1, \ldots, B_N, V$. This yields the first order, rate–distortion optimality conditions

$$\frac{1}{P_n}\frac{\partial d(\{B_n\})}{\partial B_n} = -V \text{ for all } B_n, \qquad r(\{B_n\}) = 0, \quad (4)$$

so, problem (3) can be solved by alternately updating the bit depths $B_1, \ldots, B_N$ (primal variables) and the trade-off $V$ (dual variable) until all optimality conditions are met. In other words, the optimality conditions are reached once the marginal decrease in the output distortion by assigning an infinitesimal bit is equal across layers at $-V$ and once we have assigned exactly $R$ bits per weight on average.

Since the quantization function (2) is constant a. e., a naive computation of the partial derivatives of $d$ with respect to $B_1, \ldots, B_N$ using the chain rule of differentiation does not provide a useful direction for descent. A classic result from rate–distortion theory (Gersho & Gray, 1991) is that for any random variable with finite variance, its quantization error decreases by half with every additional bit at a sufficiently high bit depth. Specifically, to our problem, one can verify that (Appendix B)

$$-\frac{1}{2\ln 2}\frac{\partial d(\{B_n\})}{\partial B_n} \approx \mathbb{E}_X \left\| \frac{\partial f(\{\Theta_n^q(B_n)\}_{n=1}^N)}{\partial \Theta_n} \Delta_n^q(B_n) \right\|^2 \quad (5)$$
$$\approx P_n H_n G_n^2 S_n^2 2^{-2B_n} \overset{\text{def}}{=} d_n(B_n)$$

in which $G_n^2$ and $S_n^2$ represent the variances of the elements

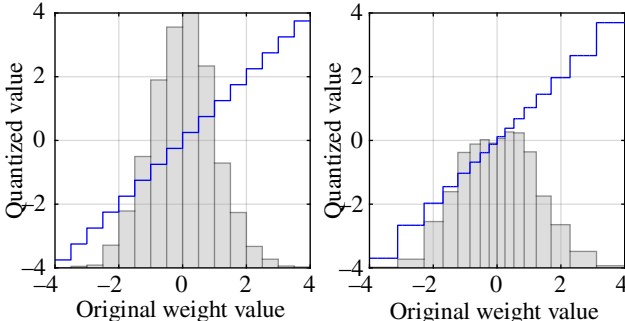

**Figure 2: Companding quantization.** Illustrated here for a 4-bit quantizer (16 quantization levels) on Gaussian weights with zero mean and unit variance. Uniform quantization of the entire range of weight values (left) leads to unduly large quantization bins (hence quantization errors) for probable weights. Companding (a sigmoid transformation) of weights to the range (0,1) prior to quantizing reduces quantization errors for more probable weights (right), which reduces output distortion.

of $\partial_{\mathbf{\Theta}_n} f(\mathbf{X}, \mathbf{\Theta}_1^q, \ldots, \mathbf{\Theta}_N^q)$, and of $\mathbf{\Theta}_n^q$, respectively, and $H_n$ is a quantization coefficient that depends only on the type of weight distribution, with $H_n = 1.42$ for Gaussian, 0.72 for Laplace, etc. (Gersho & Gray, 1991). Assuming weights are distributed across layers with $H_1 = \cdots = H_N$, factors $H_n$ and the constant $-\frac{1}{2\ln 2}$ can be removed from the above expression without affecting the solution of (3).

Coupled with the above closed-form expression (5) for the partial derivatives, optimality conditions (4) naturally lend themselves to dual ascent methods for solving (3). The idea underlying dual ascent (Boyd et al., 2011) is to alternately update the primal $B_1, \ldots, B_N$, and dual $V$ variables, with one set held fixed while updating the other variables. After initializing $B_1 = \cdots B_N = \infty$, $V$ to some small positive number, and computing $G_1^2, \ldots, G_N^2$, both sets of variables $\{B_n\}, V$ can be updated iteratively via

$$B_n \leftarrow \text{clamp}\left(\frac{1}{2}\log_2\left(\frac{G_n^2 S_n^2}{V/2\ln 2}\right), 0, B_{\max} = 8\right) \quad (6)$$
$$V \leftarrow V + \beta\left(\sum_{n=1}^N P_n B_n - \left(\sum_{n=1}^N P_n\right)R\right)$$

in which $\beta$ represents a step size for dual update. Figure 1 illustrates the optimality conditions for bit depths. With $G_n^2$ and $S_n^2$ fixed, dual ascent steps (6) converge within a few iterations (tol = $10^{-6}$ bit, step size $\beta = 2$) after which the obtained $B_n$ are rounded to integers. The non-linear nature of the least squares objective $d$ (3) means that iteration (6) needs to be repeated once bit depths $B_n$ are updated. Using the updated $B_n$, the quantized weights $\mathbf{\Theta}_n^q(B_n)$ are obtained along with the new gradient variances $G_n^2$, based on which the variables $\{B_n\}$ can be further updated via (6).

Evaluating $\partial_{\mathbf{\Theta}_n} f(\mathbf{X}, \{\mathbf{\Theta}_n^q(B_n)\}_{n=1}^N)$ across the entire set of calibration examples at every iteration can be prohibitively expensive due to the dimensions of $f(\mathbf{X}) \in \mathbb{R}^{L\times E}$ and the cost of back-propagating its elements of through $f$. We can overcome this difficulty by performing PCA on $f(\mathbf{X})$ along the embedding dimension (of $E$) and sub-sampling across

the token dimension (of $T$), and accumulating variances of gradients by back-propagating only a batch of examples at every iteration:

$$G_n^2 \leftarrow G_n^2 + \mathbb{E}_{\mathbf{X}\sim\text{batch}}\left\|\frac{\partial \mathbf{S}f(\mathbf{X}, \{\mathbf{\Theta}_n^q(B_n)\}_{n=1}^N)\mathbf{U}}{\partial \mathbf{\Theta}_n}\right\|^2, \quad (7)$$

in which $\mathbf{S}^T$ and $\mathbf{U}$ denote the PCA projection, and the sub-sampling operators, respectively. In practice, we accelerate the variance accumulation further by cycling through PCA coefficients and back-propagating only one coefficient per sample in every minibatch.

### 3.2. Quantization Step Sizes

Suppose now that weight matrices $\mathbf{\Theta}_1, \ldots, \mathbf{\Theta}_N$ need to be assigned bit depths $B_1, \ldots, B_N$ (which are not necessarily optimal yet.) We now investigate how the quantization step size $D_n$ should be decided given bit depth $B_n$. In the classic round-to-nearest scheme (RTN, Figure 2, left), $D_n$ is often chosen so that the quantizer's $2^{B_n}$ steps just cover the full range of original weight values, and $D_n$ is halved as $B_n$ is increased by one. These criteria optimize step sizes when the weights are uniformly distributed and if the objective is to minimize distortion in quantized weights.

For LLMs, the elements $\theta$ of a weight matrix can typically exhibit a light-tailed distribution $p_\theta$ (Gauss, Laplace, etc.) (Zhao et al., 2019), and partitioning the weight range into $2^{B_n}$ equal steps is sub-optimal especially at low bit depths (Cover & Thomas, 2006; Gersho & Gray, 1991). A simple alternative to non-uniform quantization using the Lloyd–Max algorithm (Lloyd, 1982; Max, 1960)—which can be computationally expensive—is "companded" quantization (Gray & Neuhoff, 1998), in which a sigmoid transform is applied to $\theta$ prior to uniform quantization to achieve finer quantization in regions of high $p_\theta$ and coarser quantization in regions of low $p_\theta$; see Figure 2 (right). When the weights $\theta$ are Laplace-distributed with mean $\mu$ and variance $S^2$, an asymptotically optimal choice of sigmoid function can be shown to be (Appendix C):

$$\sigma(\theta, S, \mu) = \frac{1 + \text{sgn}(\theta - \mu)}{2}\exp\left(-\frac{\sqrt{2}\,\text{abs}(\theta - \mu)}{3S}\right)(8)$$

that is, the normalized cubic root of a Laplace cumulative distribution function with the mean $\mu$ and variance $S^2$. The companded weights $\theta^\sigma = \sigma(\theta, S, \mu)$ can then be quantized uniformly in the range (0, 1) and communicated along with bit depth $B$, scale $S$, and mean $\mu$ for dequantization using lookup tables. In practice, $(S, \mu)$ can be treated as hyper-parameters and fine-tuned efficiently on coarse 1D grids in post-processing (Young et al., 2021) after Algorithm 1 has completed its course.

Quantization invariably causes deterministic differences to arise between the original (non-quantized) weights $\mathbf{\Theta}$ and quantized weights $\mathbf{\Theta}^q$. While these errors are customarily modeled as zero-mean noise in theoretical analyses, they are seldom zero-mean empirically, leading to a systematic

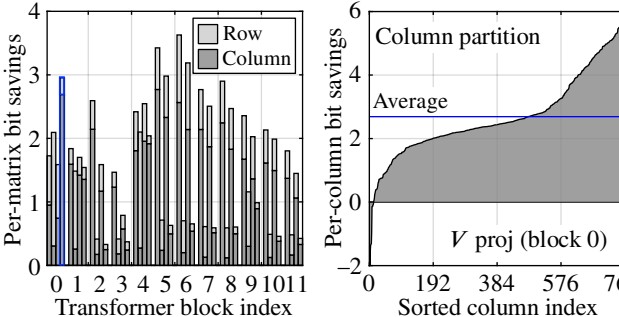

**Figure 3: Bit saving from grouping for OPT-125m.** Saving is derived by splitting each weight matrix into a collection of row or column matrices and assigning each sub-matrix its own bit depth. Savings differ across the ($Q$, $K$, $V$ and $O$) projection matrices of the model's 12 transformer blocks (left). Per- row bit saving (right—block 3, $O$-proj) can dip below zero but are always positive on average owing to Jensen's inequality.

bias in the model output and reduces prediction accuracy significantly. To compensate for the non-zero mean of the quantization errors, one can update the bias vectors for the model as $\mathbf{b}_n^q \leftarrow \mathbf{b}_n + (\mathbf{\Theta}_n^q - \mathbf{\Theta}_n)\overline{\mathbf{X}}_n$ every time the weights $\mathbf{\Theta}_n$ are quantized. Here, $\overline{\mathbf{X}}_n$ is a vector of running means of the inputs to the $n$th layer, which is accumulated during the forward pass in a manner analogous to the accumulation of $G_n^2$ during the backward pass. The corrected biases $\mathbf{b}_n^q$ are then used whenever the corresponding quantized matrices $\mathbf{\Theta}_n^q$ are used, both during gradient variance accumulation at compression time as well as inference at test time.

### 3.3. Grouping Weights

Rather than quantize at the granularity of the whole weight matrix, we can split each matrix into a collection of row or column matrices, assigning optimal bit depth and step size to each group of weights. In this case, the total number of matrices $N$ in (3) can be reinterpreted as the total number of weight groups collected across the model, and similarly reinterpret $B_n$, $D_n$ and $P_n$ as the bit-depth, step size and the number of elements of the $n$th weight group. Quantizing at the granularity of these weight groups does not increase the complexity of variance accumulation, as the same squared gradients computed via back-propagation can be averaged per weight group to produce the corresponding per-group gradient variances. To analyze the rate–distortion gain, we assume that each weight group is a column of a matrix.

For a weight matrix $\mathbf{\Theta}$ with gradient and weight variances $G^2$ and $S^2$, whose per-column variances are $G_1^2, \ldots, G_N^2$ and $S_1^2, \ldots, S_N^2$, respectively, the theoretical gain (average bit depth saving) from weight grouping can be written as

$$\gamma_{\text{group}} = \frac{1}{2}\left(\log_2\left(G^2 S^2\right) - \frac{1}{N}\sum_{n=1}^{N} \log_2\left(G_n^2 S_n^2\right)\right), \quad (9)$$

a non-negative quantity owing to Jensen's inequality. This quantity represents the bit-rate (average bit-depth) savings that can be achieved when the $n$th column is assigned $B_n = $

$\frac{1}{2}\log_2(G_n^2 S_n^2) + B$ bits for some baseline $B$, compared with assigning a uniform bit depth $B_n = \frac{1}{2}\log_2(G^2 S^2) + B$ bits across all columns when identical distribution is assumed across the $N$ columns. Figure 3 (left) plots the per-matrix bit-depth savings derived by grouping the ($Q$, $K$, $V$ and $O$) projection matrices of the OPT-125m model by its rows or columns. The sorted per-channel breakdown of the savings is also shown for the zeroth $V$ proj matrix.

In addition to the grouping of matrices into columns, one may want to sub-divide each column into a fixed number of sub-groups of weight rows to exploit the row bit saving as well. To sub-group the columns of a weight matrix $\mathbf{\Theta} \in \mathbb{R}^{N \times N}$, one can split its rows into $M$ similarly sized groups based on their total row variances: $G_1^2 S_1^2, \ldots, G_N^2 S_N^2$. By applying the same grouping to all columns of a matrix, we can signal the grouping index for each row using $\lceil \log_2 M \rceil$ bits—a negligible per-weight overhead for typical number of columns in a large weight matrix and number of groups used in practice. We illustrate partitioning and subdivision in Figure 3. In Section 4, we show the accuracy of models quantized with different numbers of row weight groups to demonstrate that the grouping mechanism used in the AWQ and GPTQ also improves rate–distortion characteristics.

## 4. Quantization Experiments

To study the rate–distortion behavior of a typical quantized LLM, we apply Algorithm 1 to the quantization of the Meta Open Pretrained Transformer (OPT) (S. Zhang et al., 2022) and Llama-2 (Touvron et al., 2023) families of language models (obtained from the HuggingFace Hub), comparing the performance of the proposed quantization method with baselines on next token prediction and question answering tasks. For calibration data, we source 128 examples from the training split of the C4 dataset (Raffel et al., 2020). We test on the test splits of WikiText2 (Merity et al., 2022) and C4 for next token prediction and those of GSM8K (Cobbe et al., 2021), ARC (Clark et al., 2018), HellaSwag (Zellers et al., 2019), PIQA (Bisk et al., 2019), and WinoGrande (Sakaguchi et al., 2021) for question-answering tasks.

**Next Token Prediction.** As our first set of experiments, we quantize the Meta OPT and Llama 2 models to 3 and 4 bits on average and measure the performance of the quantized models via perplexity, a stringent accuracy metric. We use a combined row–column group size of 512 for OPT (768 for 125M, 66B) and 256 for Llama 2 models, batch size of 16, and 17 tokens from each sequence of tokens of length 2048, and optimize for a maximum of 64 iterations. Table 1 lists the perplexity (PPL) of our quantized models on the test split of WikiText2. We select the final quantized model based on WikiText2 (best validation) but selecting the last quantized model produces similar test accuracy (within 1% of the unquantized model perplexity). For comparison, we include the perplexities of the same models quantized using round-to-nearest, GPTQ (Frantar et al., 2022), OWQ (Lee et al., 2024), AWQ (Lin et al., 2024) as well as QuIP (Chee

**Table 1: WikiText2 perplexity (test).** We quantize the Meta OPT and Llama 2 families of LLMs to 3–4 bits per weight on average via the proposed quantization method (Radio), reporting the perplexity of each quantized model on WikiText2 (test). For comparison, we also include the perplexities of models quantized using RTN, GPTQ, QuIP, OWQ, AWQ and SqueezeLLM.

| Perplexity (PPL) WikiText2 (↓) | Meta OPT (Open Pretrained Transformer) | | | | | | | | Meta Llama 2 | | |
|---|---|---|---|---|---|---|---|---|---|---|---|
| | 125M | 350M | 1.3B | 2.7B | 6.7B | 13B | 30B | 66B | 7B | 13B | 70B |
| Full Precision (FP16) | 27.65 | 22.00 | 14.63 | 12.47 | 10.86 | 10.13 | 9.56 | 9.34 | 5.47 | 4.88 | 3.32 |
| Round-to-nearest (RTN) | 37.28 | 25.94 | 48.17 | 16.92 | 12.10 | 11.32 | 10.98 | 111.36 | 5.73 | 4.98 | 3.46 |
| GPTQ (Frantar et al., 2022) | 32.05 | 23.87 | 15.47 | 12.83 | 11.14 | 10.29 | 9.57 | 9.34 | 6.07 | 5.20 | 3.59 |
| GPTQ/256 (Frantar et al., 2022) | 30.53 | 23.83 | 14.91 | 12.52 | 11.02 | 10.22 | 9.60 | 9.46 | 5.70 | 5.02 | 3.44 |
| QuIP (Chee et al., 2024) [4 bits] | 35.93 | 23.15 | 15.96 | 12.67 | 11.10 | 10.33 | 9.60 | 9.40 | 5.69 | 5.06 | 3.46 |
| OWQ (4.01 bits) (Lee et al., 2024) | 29.47 | 23.19 | 15.01 | 12.39 | 10.87 | 10.26 | 9.50 | 9.25 | 5.63 | 5.01 | 3.43 |
| AWQ/128 (Lin et al., 2024) | 29.11 | – | 14.95 | 12.74 | 10.93 | 10.22 | 9.59 | 9.39 | 5.60 | 4.97 | 3.41 |
| OmniQuant/128 (Shao et al., 2024) | 28.86 | – | 14.88 | 12.65 | 10.96 | 10.20 | 9.62 | 9.37 | 5.58 | **4.95** | 3.40 |
| SqueezeLLM (Kim et al., 2024) | – | – | 14.94 | 12.80 | 11.03 | 10.24 | 9.65 | – | 5.62 | 4.99 | 3.41 |
| Radio (4.0000 bits) (Ours) | **27.23** | 22.89 | **14.20** | **12.12** | **10.52** | **10.08** | **9.45** | **9.13** | **5.57** | 4.97 | **3.40** |
| RTN | 1.3e3 | 64.57 | 119.47 | 298.00 | 23.54 | 46.04 | 18.80 | 6.1e3 | 6.66 | 5.52 | 3.98 |
| GPTQ | 53.43 | 32.28 | 20.90 | 16.55 | 12.88 | 11.58 | 10.29 | 9.90 | 9.23 | 6.69 | 3.87 |
| GPTQ/256 | 41.22 | 29.96 | 16.98 | 13.94 | 11.39 | 10.41 | 9.81 | 11.13 | 6.75 | 5.59 | 4.00 |
| QuIP (Chee et al., 2024) [3 bits] | 34.43 | 26.02 | 17.33 | 13.84 | 12.35 | 10.57 | 9.92 | 9.46 | 6.36 | 5.50 | 3.86 |
| OWQ (3.01 bits) (Lee et al., 2024) | 35.26 | 26.59 | 16.40 | 13.21 | 11.21 | 11.48 | 9.59 | 9.28 | 6.21 | 5.36 | 3.77 |
| AWQ/128 | 36.77 | – | 16.32 | 13.54 | 11.41 | 10.67 | 9.85 | 9.63 | 6.24 | 5.32 | 3.74 |
| OmniQuant/128 (Shao et al., 2024) | 32.25 | – | 15.71 | 13.18 | 11.27 | 10.47 | 9.79 | 9.53 | **6.03** | 5.28 | 3.78 |
| SqueezeLLM (Kim et al., 2024) | – | – | 16.30 | 13.85 | 11.70 | 11.76 | 10.17 | – | 6.18 | 5.36 | 3.77 |
| Radio (3.0000 bits) (Ours) | **30.71** | **25.94** | **14.83** | **12.42** | **11.07** | **10.28** | **9.56** | **9.24** | 6.04 | **5.25** | **3.72** |

**Table 2: Effect of hyperparameters on quantized model accuracy.** Quantized model accuracy is relatively insensitive to the minibatch size (a) and number of tokens per sequence (b) used for the optimization. Smaller groups can improve quantized model accuracy at low average bit depths (c).

(a) Minibatch size and PPL

| PPL C4 (↓) | OPT (4 bits) | | OPT (3 bits) | |
|---|---|---|---|---|
| | 1.3B | 13B | 1.3B | 13B |
| FP16 | 16.07 | 12.06 | 16.07 | 12.06 |
| 2 | 16.24 | 12.12 | 16.94 | 12.36 |
| 4 | 16.24 | 12.12 | 16.94 | 12.35 |
| 8 | 16.25 | 12.11 | 16.90 | 12.34 |
| **16** | **16.22** | **12.11** | **16.86** | **12.32** |
| 32 | 16.24 | 12.12 | 16.88 | 12.36 |

(Batch size)

(b) Number of tokens and PPL

| PPL C4 (↓) | OPT (4 bits) | | OPT (3 bits) | |
|---|---|---|---|---|
| | 1.3B | 13B | 1.3B | 13B |
| FP16 | 16.07 | 12.06 | 16.07 | 12.06 |
| 3 | 16.40 | 12.29 | 17.05 | 12.47 |
| 5 | 16.28 | 12.18 | 16.93 | 12.37 |
| 9 | 16.24 | 12.12 | 16.91 | 12.35 |
| **17** | **16.22** | **12.11** | **16.86** | **12.32** |
| 33 | 16.21 | 12.10 | 16.87 | 12.34 |

(Num tokens)

(c) Group size and PPL

| PPL C4 (↓) | OPT (4 bits) | | OPT (3 bits) | |
|---|---|---|---|---|
| | 1.3B | 13B | 1.3B | 13B |
| FP16 | 16.07 | 12.06 | 16.07 | 12.06 |
| 64 | 16.16 | 12.10 | 16.62 | 12.26 |
| 128 | 16.17 | 12.10 | 16.70 | 12.29 |
| 256 | 16.20 | 12.10 | 16.77 | 12.32 |
| **512** | **16.22** | **12.11** | **16.86** | **12.32** |
| 1024 | 16.23 | 12.11 | 16.99 | 12.42 |

(group size)

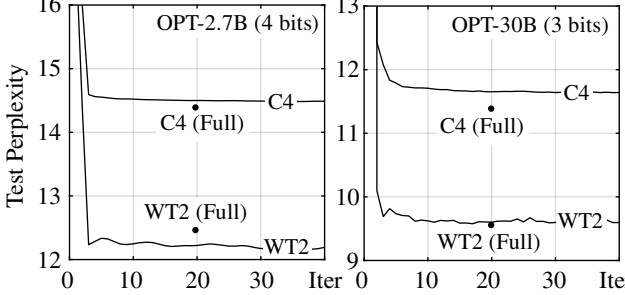

**Figure 4: Perplexity across optimization iterations.** Calibrated on C4 (train) using a batch size of 16. Row groups of size 512 used. Perplexity decreases rapidly across the first thirty iterations, after which iterations can be early terminated. Note monotonicity for C4 (test), whose distribution is more like the calibration data.

et al., 2024) based on the code provided by the respective authors; see Appendix D for details. Relative to the next best performing methods, the proposed method provides a perplexity reduction of up to 4.55 for the 3-bit OPT-125M model but minor perplexity gains (0.00–0.01) are observed for the 3-bit OPT-66B and Llama 2 70B models. In this comparison, AWQ uses a group size of 128, incurring 2–4 times as many overhead bits as the proposed method, and OWQ, by its nature, operates at average bit depths that are 0.01–0.05 bits higher per weight on average than proposed.

**Hyperparameters/Ablations.** To analyze the influence of Radio hyperparameters on the accuracy of the quantized models, we quantize the OPT-1.3B and -13B models over a range of minibatch sizes and token counts (optimization hyperparameters) as well as the group size (quantization hyperparameter), with each hyperparameter varied while keeping the others fixed at their optimized values. (The optimal hyperparameter values are batch size: 16, token count: 17, and group size: 512.) The perplexity of the quantized models is then measured on the C4 test data. Table 2 (a–b) demonstrates that Radio is largely insensitive to the values of optimization hyperparameters over a wide range. From Table 2 (c), we see that smaller group sizes generally improve the performance of the quantized models at lower average bit depths, but this also leads to

**Table 3: Ablations and pruning effects of quantization.** A small fraction of weights is quantized to zero and pruned away due to low variance, with smaller groups increasing the degree of pruning (a). Quantization incurs overhead bits for signaling group indices and location and scale parameters of groups (b).

(a) Ablations (mixed precision and step sizes)

| Perplexity C4 ($\downarrow$) | OPT (4 bits) | | OPT (3 bits) | |
|---|---|---|---|---|
| | 1.3B | 13B | 1.3B | 13B |
| RTN (Round-To-Nearest) | 24.51 | 13.36 | 4.2e3 | 3.2e3 |
| + MMSE Step Sizes | 16.98 | 12.26 | 21.64 | 13.34 |
| + Mixed Precision Depths | 16.29 | 12.12 | 18.48 | 12.73 |
| + Companding | 16.22 | 12.11 | 16.86 | 12.32 |
| = Radio (Ours) | **16.22** | **12.11** | **16.86** | **12.32** |

(b) Pruned columns (%)

| | Pruned (%) | Meta OPT (4 bits) | | |
|---|---|---|---|---|
| | | 350M | 1.3B | 13B |
| Group size | 64 | 0.57 | 2.13 | 2.18 |
| | 128 | 0.61 | 2.19 | 2.31 |
| | 256 | 0.67 | 2.10 | 2.16 |
| | **512** | **0.68** | **2.07** | **2.00** |
| | 1024 | 0.68 | 2.08 | 1.92 |

(c) Overhead bits (%)

| | Overhead bits (%) | Meta OPT (4 bits) | | | |
|---|---|---|---|---|---|
| | | 350M | 1.3B | 13B | 30B |
| Group size | 64 | 10.33 | 10.30 | 10.28 | 13.70 |
| | 128 | 5.18 | 5.16 | 5.15 | 6.86 |
| | 256 | 2.60 | 2.59 | 2.58 | 3.44 |
| | **512** | **1.30** | **1.30** | **1.30** | **1.72** |
| | 1024 | 0.64 | 0.65 | 0.65 | 0.86 |

**Table 4: 2.x-bit quantization and downstream tasks.** Quantized to 2.x bits per weight, Radio reduces perplexity considerably compared with OWQ models quantized to the same (a). Quantized model accuracy measured by performance on tasks such as GSM8K (b). Group size of 256 is used.

(a) Perplexity of 2.1–2.8-bit Llama 2 models

| Perplexity WikiText2 ($\downarrow$) | Llama 2 7B (2.1–2.8 bits) | | | | |
|---|---|---|---|---|---|
| | 2.1 | 2.2 | 2.4 | 2.6 | 2.8 |
| FP16 (Full Precision) | 5.47 | 5.47 | 5.47 | 5.47 | 5.47 |
| OWQ (Lee et al., 2024) | 39.56 | 11.25 | 10.79 | 10.43 | 10.24 |
| OWQ/256 | 10.34 | 10.01 | 9.98 | 9.50 | 9.26 |
| OWQ/128 | 10.01 | 9.66 | 9.42 | 9.38 | 9.14 |
| Radio/256 (Ours) | **9.47** | **8.39** | **7.05** | **6.56** | **6.21** |

(b) Scores for 3-bit Llama-2 models on GSM8K and QA

| Score (%) Llama-2 ($\uparrow$) | GSM8K | | | Average QA | | |
|---|---|---|---|---|---|---|
| | 7B | 13B | 70B | 7B | 13B | 70B |
| FP16 (Full Precision) | 64.83 | 67.82 | 72.36 | 14.10 | 23.43 | 53.90 |
| RTN (Round-To-Nearest) | 1.82 | 1.67 | 6.14 | 39.32 | 52.15 | 58.22 |
| GPTQ/256 (Frantar et al., 2022) | 6.60 | 14.48 | 46.47 | 61.40 | 64.94 | 70.58 |
| AWQ/256 (Lin et al., 2024) | 6.97 | 16.76 | 48.07 | 62.48 | 65.95 | 71.29 |
| Radio/256 (3.0000 bits) (Ours) | **7.81** | **18.20** | **49.81** | **62.82** | **66.37** | **71.87** |

(c) Scores for 3-bit Llama-2 models on common sense QA (Arc-Challenging, Arc-Easy, HellaSwag, PIQA, Winogrande)

| Score (%) Llama-2 ($\uparrow$) | Arc (Challenge) | | | Arc (Easy) | | | HellaSwag | | | PIQA | | | Winogrande | | |
|---|---|---|---|---|---|---|---|---|---|---|---|---|---|---|---|
| | 7B | 13B | 70B | 7B | 13B | 70B | 7B | 13B | 70B | 7B | 13B | 70B | 7B | 13B | 70B |
| FP16 (Full Precision) | 43.34 | 48.38 | 54.27 | 76.30 | 79.42 | 82.74 | 57.13 | 60.04 | 64.76 | 78.07 | 79.05 | 82.15 | 69.30 | 72.22 | 77.90 |
| RTN (Round-To-Nearest) | 20.73 | 30.63 | 37.20 | 34.97 | 60.65 | 65.66 | 31.09 | 43.73 | 51.01 | 57.34 | 70.40 | 73.12 | 52.49 | 55.33 | 64.09 |
| GPTQ/256 (Frantar et al., 2022) | 38.23 | 43.34 | 52.13 | 72.26 | 76.64 | 80.85 | 53.02 | 57.65 | 62.60 | 75.63 | 77.48 | 80.52 | 67.88 | 69.61 | 76.80 |
| QuIP/256 (Chee et al., 2024) | 38.74 | 45.39 | 52.30 | 72.77 | – | 80.68 | **54.08** | 76.77 | 63.04 | 77.09 | **78.78** | 81.12 | 66.85 | 70.40 | 76.72 |
| AWQ/256 (Lin et al., 2024) | 41.13 | 45.05 | 53.24 | **73.36** | **77.95** | 81.69 | 54.06 | 57.83 | 63.64 | 75.84 | 77.26 | 81.66 | 68.03 | **71.67** | 76.24 |
| Radio/256 (3.0000 bits) (Ours) | **41.21** | **45.73** | **53.84** | 72.60 | **77.95** | **82.32** | 53.95 | **58.55** | **63.86** | **77.20** | 78.51 | **81.88** | **69.14** | 71.11 | **77.43** |

higher overheads (discussed later). Figure 4 plots quantized model accuracy across optimization iterations in the case where baseline hyperparameter values are used. It appears that about 20 iterations are needed for quantization parameters (bit depth decisions) to reach their optima. Table 3 (a) shows ablations of our quantized OPT models by starting with RTN and adding different components (Jeon et al., 2023). See Table 5 for C4 perplexity results.

**Pruning Due to Quantization.** Our method quantizes very low-variance weights of weight matrices to zero and effects a form of weight pruning, which has been demonstrated to improve test accuracy (Hassibi & Stork, 1992). Table 3 (b) lists the percentages of zero-quantized weights in the OPT-1.3B and 13B models quantized to 3 and 4 bits per weight on average. We observe that using smaller grouping sizes increases the number of pruned weights since this enables lower-variance weights in each column to be grouped together and quantized to zero. However, smaller groups lead to higher overheads so that small improvements in generalization due to pruning come at the cost of signaling the overhead bits. Table 3 (c) lists the number of overhead bits (group indices and FP16 encodings of the location and scale parameters of each weight group) as a percentage of

the total quantized weight bits. These overheads are in line with those of other algorithms which must similarly signal zero points and step sizes of the quantization grid (Lee et al., 2024).

**2.x-bit Llama-2.** We study the accuracy of Llama 2 models quantized to 2.x bits using Radio and OWQ, both of which are capable of quantizing models to fractional average bit depths. To allow a more comprehensive study, we compare against OWQ with no grouping, as well as with group sizes of 128 and 256. We can see from Table 4 (a) that Radio-quantized Llama-2 models are considerably more accurate at these bit depths than their OWQ counterparts. This is expected since Radio assigns bit depths from the range $(0, B_{max})$ commensurately with gradient variances whereas OWQ opts to preserve the most sensitive weights in FP16 and quantize the rest to 2 bits (Lee et al., 2024). In terms of execution time, Radio (for 64 iterations) and OWQ/GPTQ require 47 minutes and 18 minutes, respectively (excluding testing), to quantize the 7B model on an Nvidia A100.

**Downstream Tasks (Common Sense QA, GSM8K).** To show the impact of model quantization on downstream tasks, we list in Table 4 (b–c) the accuracy of our quantized

**Table 5: C4 perplexity (validation).** We quantize the Meta OPT and Llama 2 families of LLMs to 3–4 bits per weight on average using the proposed quantization method, reporting the perplexity of each quantized model on the C4 dataset. For comparison, we also include the perplexities of models quantized using RTN, GPTQ, QuIP, OWQ, and AWQ.

| Perplexity (PPL) C4 (↓) | Meta OPT (Open Pretrained Transformer) | | | | | | | | Meta Llama 2 | | |
|---|---|---|---|---|---|---|---|---|---|---|---|
| | 125M | 350M | 1.3B | 2.7B | 6.7B | 13B | 30B | 66B | 7B | 13B | 70B |
| Full Precision (FP16) | 26.56 | 22.59 | 16.07 | 14.34 | 12.71 | 12.06 | 11.44 | 10.99 | 6.97 | 6.46 | 5.52 |
| **4 bits** | | | | | | | | | | | |
| Round-to-nearest (RTN) | 33.91 | 16.21 | 24.51 | 18.43 | 14.36 | 13.36 | 13.46 | 283.31 | 7.86 | 7.16 | 6.01 |
| GPTQ (Frantar et al., 2022) | 29.42 | 24.14 | 16.73 | 14.85 | 12.99 | 12.24 | 11.56 | 11.08 | 7.86 | 7.06 | 5.90 |
| GPTQ/256 (Frantar et al., 2022) | 28.36 | 24.18 | 16.47 | 14.64 | 12.88 | 12.15 | 11.50 | 11.12 | 7.58 | 6.88 | 5.79 |
| QuIP (Chee et al., 2024) | 27.85 | 23.39 | 17.20 | 14.58 | 12.87 | 12.17 | 11.51 | 11.03 | 7.57 | 6.91 | 5.80 |
| OWQ (4.01 bits) (Lee et al., 2024) | 27.93 | 23.37 | 16.49 | 14.60 | 12.83 | 12.17 | 11.49 | 11.02 | 7.59 | 6.94 | 5.81 |
| AWQ/128 (Lin et al., 2024) | 27.79 | – | 16.42 | 14.58 | 12.84 | 12.15 | 11.50 | 11.04 | 7.44 | 6.84 | 5.77 |
| OmniQuant/128 (Shao et al., 2024) | 27.59 | – | 16.38 | 14.56 | 12.82 | 12.16 | 11.50 | 11.04 | **7.12** | **6.56** | **5.58** |
| SqueezeLLM (Kim et al., 2024) | – | – | 16.36 | 14.55 | 12.82 | 12.15 | 11.50 | – | 7.12 | 6.57 | 5.58 |
| Radio (4.0000 bits) (Ours) | **27.27** | **23.20** | **16.24** | **14.44** | **12.79** | **12.11** | **11.48** | **11.01** | 7.40 | 6.83 | 5.76 |
| **3 bits** | | | | | | | | | | | |
| RTN | 839.97 | 55.96 | 4.2e3 | 1.1e4 | 4.4e3 | 3.2e3 | 1.1e3 | 3.5e3 | 521.22 | 14.01 | 11.06 |
| GPTQ | 42.64 | 29.90 | 20.46 | 17.48 | 14.56 | 13.16 | 12.14 | 11.53 | 11.44 | 8.98 | 7.12 |
| GPTQ/256 (Frantar et al., 2022) | 35.00 | 28.84 | 18.07 | 15.84 | 13.50 | 12.57 | 11.78 | 12.29 | 8.92 | 7.65 | 6.21 |
| QuIP (Chee et al., 2024) | 31.37 | **25.58** | 18.15 | 15.92 | 13.66 | 12.40 | 11.67 | **11.16** | 8.48 | 7.49 | 6.10 |
| OWQ (3.01 bits) (Lee et al., 2024) | 31.28 | 26.40 | 17.69 | 15.36 | 13.23 | 13.29 | 11.69 | 11.17 | 8.59 | 7.65 | 6.16 |
| AWQ/128 | 32.91 | – | 17.81 | 15.49 | 13.34 | 12.55 | 11.75 | 11.26 | 8.30 | 7.31 | 6.04 |
| OmniQuant/128 (Shao et al., 2024) | 31.30 | – | 17.46 | 15.33 | 13.28 | 12.50 | 11.73 | 11.22 | 7.75 | 6.98 | 5.85 |
| SqueezeLLM (Kim et al., 2024) | – | – | 17.19 | 15.62 | 13.41 | 13.55 | 11.85 | – | **7.72** | **6.97** | **5.73** |
| Radio (3.0000 bits) (Ours) | **30.05** | 26.20 | **16.88** | **14.91** | **13.14** | **12.35** | **11.62** | 11.19 | 8.04 | 7.22 | 5.99 |

Llama-2 models on the ARC-C, -E (Clark et al., 2018), HellaSwag (Zellers et al., 2019), PIQA (Bisk et al., 2019) and Winogrande (Sakaguchi et al., 2021) common sense question answering, and GSM8K (Cobbe et al., 2021) math problem solving tasks. We set our group size and the group size of GPTQ and AWQ to 256. We observe that Radio produces slightly higher scores than the GPTQ and AWQ quantized 3-bit models while RTN leads to severely diminished scores despite having similar perplexity scores as Radio on WikiText2 (Table 1). We include example responses to GSM8K questions produced by different 3-bit quantized Llama-2-70B models in Appendix E.

**Timing results.** To show the running time behavior of the proposed quantization method, Table 6 lists running times of Radio on Llama 2 models of different sizes, measured on Nvidia A100. It can be seen that Radio's running time grows close to linearly with model size. Table 7 lists the acceleration achieved by our custom quantized matrix-vector multiply kernel, with acceleration ranging between 1.4 and 3.3 depending on the embedding dimensionality.

## 5. Discussion

Formulating weight quantization as a convex optimization problem as we have done here yields two benefits. First, it explicates the objective we seek to optimize (minimizing output distortion in this case) and sets us on a path to solve the right problem using modern automatic differentiation tools e.g. PyTorch's autograd library. Second, it enables us to interpret many earlier Hessian-based methods (Frantar et al., 2022; Lee et al., 2024; Dong et al., 2019; Chen et al., 2021) as heuristics for approximate optimization of the true

**Table 6: Radio running times.** We quantize the Meta Llama 2 family of LLMs to ~3 bits per weight on average and measure the running time of the proposed method. We also include the running times of GPTQ, QuIP, OWQ, AWQ, and SqueezeLLM.

| Quantization runtimes with 128 examples from C4 (seq length 2048) | Meta Llama 2 | | |
|---|---|---|---|
| | 7B | 13B | 70B |
| Full Precision (FP16) | 0m | 0m | 0m |
| **3 bits** | | | |
| RTN | <1m | <1m | 5m |
| GPTQ/256 (Frantar et al., 2022) | 10m | 18m | 90m |
| QuIP (Chee et al., 2024) | 36m | 80m | 9h |
| OWQ (3.01 bits) (Lee et al., 2024) | 28m | 50m | 4h |
| AWQ/128 | **7m** | **17m** | **87m** |
| OmniQuant/128 (Shao et al., 2024) | 56m | 132m | 15h |
| SqueezeLLM (Kim et al., 2024) | 11m | 23m | 92m |
| Radio (3.0000 bits) (Ours) | 47m | 97m | 11h |

underlying quantization objective. Note (2) is a non-linear system of equations in the bit depth variables, so that any non-iterative solution is necessarily only an approximate one if one's goal is to optimize an objective like (2). Recent high-performance model quantization techniques (Chee et al., 2024; Frantar et al., 2022; Frantar & Alistarh, 2022; Lee et al., 2024) ultimately trace their lineage back to the classic Optimal Brain Surgeon (OBS) algorithm (Hassibi & Stork, 1992), which is a convex formulation for weight pruning, not quantization (Appendix F). As a result, these methods inherit the need for fine-tuning weights as part of the quantization process, making them less suitable for the quantization of activations at inference time, where fine-tuning would lead to unacceptable delays in the inference pipeline.

Our experimental results demonstrate that a more accurate

Table 7: Acceleration due to quantized mat-vector multiplies relative to multiplication in FP16. A 3-bit weight matrix of dimension $N \times M$ multiplies a vector of length $M$ to produce a vector of length $N$ (denoted $M \to N$).

| Model (Embedding) | Acceleration factors | | | |
|---|---|---|---|---|
| | $E{\to}E$ | $E{\to}4E$ | $4E{\to}E$ | Overall |
| OPT-1.3B ($E = 1024$) | 0.9 | 2.1 | 2.7 | 1.4 |
| OPT-1.3B ($E = 2048$) | 2.7 | 2.3 | 2.5 | 2.7 |
| OPT-6.7B ($E = 4096$) | 2.4 | 3.1 | 3.1 | 2.8 |
| OPT-30B ($E = 7168$) | **3.2** | **3.2** | **3.1** | **3.2** |
| OPT-66B ($E = 9216$) | 3.2 | 3.1 | 3.2 | 3.2 |
| OPT-175B ($E = 12288$) | 3.2 | 3.2 | 3.8 | 3.3 |

(3 bits)

characterization of the LLM quantization problem can lead to better compression outcomes. While the smaller OPT-125M model is too limited for practical usefulness in most situations, its relative incompressibility helps contrast the performance of the different weight quantization methods themselves (Table 1). With larger models like OPT-66B and Llama 2-66B, most quantization techniques (including RTN) perform similarly, suggesting that larger language models are more compressible generally. At first glance, a RTN may seem sufficient for quantizing larger models, but upon a closer look, RTN-quantized models lead to severely reduced accuracy on downstream tasks such as GSM8K (Table 4 (a)), which highlights the importance of validating the accuracy of quantized models across multiple tasks and datasets (Jaiswal et al., 2024). Interestingly, increasing the number of calibration examples (from 128 to 1024) did not noticeably affect the quantized model's perplexity on C4 (± 0.01), which agrees with findings from previous works; see (Kim et al., 2024; Hubara et al., 2021). We will discuss the quantization of activations and more advanced methods for model quantization in the sequel.

Finally, our CUDA matmul kernel (Appendix A) provides acceleration for matrix-vector multiplies by dequantizing mixed precision weights to a floating-point representation (FP16) dynamically then multiplying them by activations of the same representation. For the 12288 × 49152 weight matrix of Meta's OPT-175B quantized to 3 bits per weight on average, our custom CUDA kernel leads to a 3.8x speed up over the FP16 matrix-vector multiply performed using the default cuBLAS matmul on Nvidia A6000. Accelerated matrix-vector multiply and low quantization complexity of our quantization approach allows us to apply Radio also to the quantization of batched activations, where quantization efficiency becomes paramount.

## 6. Conclusion

Here, we showed that a rate–distortion framework can lead to better LLM quantization outcomes. Despite numerous advances in methods for LLM compression, there has not been an extensive study in the rate–distortion-theory aspects of model quantization and optimization techniques required to solve the resulting optimization problem. This work fills the gap in the current literature by introducing the rate–distortion framework and a stochastic numerical solver for the rate–distortion optimization problem.

## Reproducibility Statement

To ensure the reproducibility of results in this work, we make our PyTorch Radio program available on our GitHub project website, where readers can also ask questions about this work. Appendix A lists our CUDA kernel. Appendices B–C provide derivations for our main theoretical results and Appendix D additionally details the PyTorch code and command line options used to obtain the results of GPTQ (Frantar et al., 2022), OWQ (Lee et al., 2024), and AWQ (Lin et al., 2024).

## Impact Statement

This paper aims to advance the theory of rate and distortion for LLM compression. There are many potential societal consequences of our work, none which we feel needs to be specifically highlighted here.

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

## A. Radio Kernel for Matrix-Vector Multiplication

For completeness, we provide here a reference implementation for multiplication between a mixed-precision quantized matrix and full-precision vector multiplication. Here, we assign a single bit depth to each group of 4 rows, leading to e.g. 12288 different bit depths in the case of the $49152 \times 12288$ weight matrix (MLP layer) of the OPT-175B model. Consider a thread block size of $256 \times 256$. Each of $256 \times 256$ block, in turn, entails $1 \times 256$ threads, with each thread dequantizing a $256 \times 1$ column of weights and multiplying them with the matching $1 \times 256$ segment of a vector input. Bit depth changes every 4 rows but every thread will go through the same bit depth change in the course of multiplication, allowing divergence-free (and uniform memory access) operations.

```
__constant__ float lutable[256] = { DEQUANT }; // dequantized values defined in macros.h

template <typename scalar_t>
__global__ void VecQuant3MatMulKernel(
    const  scalar_t* __restrict__  vec,
    const       int* __restrict__  mat,
          scalar_t* __restrict__  mul,
    const   uint8_t* __restrict__  depths,
    const  scalar_t* __restrict__  scales,
    const       int* __restrict__  i_s,
    const   uint8_t* __restrict__  shifts,

    int height,
    int width) {
  int row = BLOCKHEIGHT * blockIdx.x;
  int col =  BLOCKWIDTH * blockIdx.y + threadIdx.x;

  __shared__ scalar_t blockvec[BLOCKWIDTH];
  __shared__ scalar_t lut[BLOCKWIDTH];

  blockvec[threadIdx.x] = scales[threadIdx.x / 4] * vec[(row / BLOCKHEIGHT) * BLOCKWIDTH +
threadIdx.x];
  lut[threadIdx.x] = lutable[threadIdx.x];
  __syncthreads();

  scalar_t res = 0;
  int i = i_s[blockIdx.x * gridDim.y + blockIdx.y] + threadIdx.x;
  // int i = width * row + col;
  int shift = shifts[blockIdx.x * gridDim.y + blockIdx.y];

  uint64_t tmp_curr;
  uint32_t tmp_read;
  uint32_t depth_;

  int j = 0, k = 0;

  tmp_read = reinterpret_cast<const uint32_t*>(mat)[i];
  tmp_curr = static_cast<uint64_t>(tmp_read) << 32;
  shift += 32;
  i += width;

  while (k < BLOCKWIDTH) {
    depth_ = reinterpret_cast<const uint32_t*>(depths)[j];

    int depth, bmask;
    uint32_t index;
    scalar_t szero, *table;
    for (int d = 0; d < 32; d += 8) { // for each of the 4 depth groups (represented in 8
bits)
      depth = (depth_ >> (d + 0)) &  7;
      bmask = (1 << depth) - 1;
```

```
        szero = (static_cast<int>((depth_ >> (d + 3)) & 31) - 16) * 0.03125f;
        table = reinterpret_cast<scalar_t*>(lut + (1 << depth));

        if (shift + 4 * depth > 64) { // will run out of bits, read more
          tmp_read = reinterpret_cast<const uint32_t*>(mat)[i];
          tmp_curr       =       static_cast<uint64_t>(tmp_read)       <<       32       |
static_cast<uint64_t>(tmp_curr) >> 32;
          shift -= 32;
          i += width;
        }
        index = (static_cast<uint32_t>(tmp_curr >> shift) & bmask);
        res += blockvec[k + 0] * (szero + table[index]);
        shift += depth;
        index = (static_cast<uint32_t>(tmp_curr >> shift) & bmask);
        res += blockvec[k + 1] * (szero + table[index]);
        shift += depth;
        index = (static_cast<uint32_t>(tmp_curr >> shift) & bmask);
        res += blockvec[k + 2] * (szero + table[index]);
        shift += depth;
        index = (static_cast<uint32_t>(tmp_curr >> shift) & bmask);
        res += blockvec[k + 3] * (szero + table[index]);
        shift += depth;

        k += 4;
      }
      j += 1;
    }
  atomicAdd(&mul[col], res);
}
```

## B. Derivation of Equation (5)

To derive our main equation (5), we appeal to a linearized relationship between model weights and output, as well as standard results from rate–distortion theory (Gersho & Gray, 1991) that relate the quantization error of a random source to output distortion at a high bit depth, where the linearized model relationship is a good approximation. Let us start with our quantization objective

$$d(B_1, B_2, \ldots, B_N) = \mathbb{E}_{\mathbf{X}} \| f(\mathbf{X}, \boldsymbol{\Theta}_1^q(B_1), \boldsymbol{\Theta}_2^q(B_2), \ldots, \boldsymbol{\Theta}_N^q(B_N)) - f(\mathbf{X}) \|_F^2, \tag{10}$$

in which $f(\mathbf{X}) = f(\mathbf{X}, \boldsymbol{\Theta}_1(B_1), \boldsymbol{\Theta}_2(B_2), \ldots, \boldsymbol{\Theta}_N(B_N))$ denotes the output of the unquantized model given input $\mathbf{X}$. We can write the residual and Jacobian of $f$ at $(\mathbf{X}, \boldsymbol{\Theta}_1^q(B_1), \boldsymbol{\Theta}_2^q(B_2), \ldots, \boldsymbol{\Theta}_N^q(B_N))$ as

$$r(\mathbf{X}, \boldsymbol{\Theta}_1^q, \boldsymbol{\Theta}_2^q, \ldots, \boldsymbol{\Theta}_N^q) = (r_1, \ldots, r_M)(\mathbf{X}, \boldsymbol{\Theta}_1^q, \boldsymbol{\Theta}_2^q, \ldots, \boldsymbol{\Theta}_N^q) = f(\mathbf{X}, \boldsymbol{\Theta}_1^q, \boldsymbol{\Theta}_2^q, \ldots, \boldsymbol{\Theta}_N^q) - f(\mathbf{X})$$

$$J(\mathbf{X}, \boldsymbol{\Theta}_1^q, \boldsymbol{\Theta}_2^q, \ldots, \boldsymbol{\Theta}_N^q) = \left( \frac{\partial f(\mathbf{X}, \boldsymbol{\Theta}_1^q, \ldots, \boldsymbol{\Theta}_N^q)}{\partial \boldsymbol{\Theta}_1}, \frac{\partial f(\mathbf{X}, \boldsymbol{\Theta}_1^q, \ldots, \boldsymbol{\Theta}_N^q)}{\partial \boldsymbol{\Theta}_2}, \ldots, \frac{\partial f(\mathbf{X}, \boldsymbol{\Theta}_1^q, \ldots, \boldsymbol{\Theta}_N^q)}{\partial \boldsymbol{\Theta}_N} \right) \tag{11}$$

and proceed to write the gradient and Hessian of the objective (10) in terms of the $r$ and $J$ above as

$$\nabla d(\mathbf{X}, \boldsymbol{\Theta}_1^q, \boldsymbol{\Theta}_2^q, \ldots, \boldsymbol{\Theta}_N^q) = (J^T r)(\mathbf{X}, \boldsymbol{\Theta}_1^q, \boldsymbol{\Theta}_2^q, \ldots, \boldsymbol{\Theta}_N^q)$$

$$\nabla^2 d(\mathbf{X}, \boldsymbol{\Theta}_1^q, \boldsymbol{\Theta}_2^q, \ldots, \boldsymbol{\Theta}_N^q) = (J^T J)(\mathbf{X}, \boldsymbol{\Theta}_1^q, \boldsymbol{\Theta}_2^q, \ldots, \boldsymbol{\Theta}_N^q) + \underbrace{\sum_{m=1}^{M} (r_m \nabla^2 r_m)(\mathbf{X}, \boldsymbol{\Theta}_1^q, \boldsymbol{\Theta}_2^q, \ldots, \boldsymbol{\Theta}_N^q)}_{\approx 0} \tag{12}$$

in which the second term of $\nabla^2 d$ is approximately zero either because the residuals $r_m$ are relatively small, or they are close to affine in $(\boldsymbol{\Delta}_1^q, \boldsymbol{\Delta}_2^q, \ldots, \boldsymbol{\Delta}_N^q)$ so that $\nabla^2 r_m$ are relatively small, which is the case in the vicinity of the solution.

Using (12), we can now express the local quadratic approximation of (10) about $(B_1, \ldots, B_N)$ as

$$\hat{d}(B_1, \ldots, B_N) \overset{(a)}{=} \mathbb{E}_{\mathbf{X}} \left[ \left( \boldsymbol{\Delta}_1^q(B_1), \ldots, \boldsymbol{\Delta}_N^q(B_N) \right) \left( (J^T J)(\mathbf{X}, \boldsymbol{\Theta}_1^q, \ldots, \boldsymbol{\Theta}_N^q) \right) \left( \boldsymbol{\Delta}_1^q(B_1), \ldots, \boldsymbol{\Delta}_N^q(B_N) \right)^T \right]$$

$$+ \underbrace{\mathbb{E}_{\mathbf{X}} \left[ \left( \Delta_1(B_1), \ldots, \Delta_N(B_N) \right)^T \left( (J^T r)(\mathbf{X}, \boldsymbol{\Theta}_1^q, \boldsymbol{\Theta}_2^q, \ldots, \boldsymbol{\Theta}_N^q) \right) \right]}_{= 0} \tag{13}$$

$$\overset{(b)}{=} \sum_{n=1}^{N} \mathbb{E}_{\mathbf{X}} \left[ (J^T J)_{nn}(\mathbf{X}, \boldsymbol{\Theta}_1^q, \ldots, \boldsymbol{\Theta}_N^q) \right] \mathbb{E} \left[ \Delta_n^2(B_n) \right] \overset{(c)}{=} \sum_{n=1}^{N} \underbrace{P_n G_n^2 H_n S_n^2 2^{-2B_n}}_{= d_n(B_n)}$$

in which the zero expectation of the linear term in (a) follows from the zero means of quantization errors $\Delta_1, \ldots, \Delta_N$, (b) follows from the uncorrelatedness of $\Delta_1, \ldots, \Delta_N$, and (c) follows from our definition of gradient variance $G_n^2 = P_n^{-1} \mathbb{E}_{\mathbf{X}} \left[ (J^T J)_{nn}(\mathbf{X}, \boldsymbol{\Theta}_1^q, \ldots, \boldsymbol{\Theta}_N^q) \right]$ together with the result from rate–distortion theory (Gersho & Gray, 1991) that relates the variance of random quantization error $\mathbb{E} \left[ \Delta_n^2(B_n) \right] = H_n S_n^2 2^{-2B_n}$ to the variance $S_n^2$ of the random source, and the coefficient $H_n$, and bit depth $B_n$ of quantization. Expression (5) for the partial derivatives of $d$ with respect to $B_n$ follows directly from the properties of the derivative of an exponential.

Since (10) is a non-linear least-squares objective and its gradient depends on the gradient variances $G_1^2, G_2^2, \ldots, G_N^2$, its minimization requires an iterative update of $\boldsymbol{\Theta}_1^q, \boldsymbol{\Theta}_2^q, \ldots, \boldsymbol{\Theta}_N^q$ via the choice of $B_1, B_2, \ldots, B_N$ and re-evaluation of the gradient variances $G_1^2, G_2^2, \ldots, G_N^2$ at $\boldsymbol{\Theta}_1^q, \boldsymbol{\Theta}_2^q, \ldots, \boldsymbol{\Theta}_N^q$. This is similar to the local Hessian evaluated by the Gauss–Newton method (Nocedal & Wright, 2006) every time the descent direction is re-computed. One can think of $G_1^2, G_2^2, \ldots, G_N^2$ as the diagonal elements of a non-diagonal Hessian matrix used in e.g. the Gauss–Newton method, but whose off-diagonal elements disappear in the expectation due to multiplication by uncorrelated quantization errors $\boldsymbol{\Delta}_1^q, \ldots, \boldsymbol{\Delta}_N^q$.

## C. Derivation of Equation (8)

To derive our sigmoid companding function (8), we turn to results from rate–distortion theory that relate the mean square error of quantization of weights $\theta$ to the density $p_\theta$ of $\theta$ and the density $\lambda(\theta)$ of quantization levels, where $2^B \int_a^b \lambda(\theta) \, d\theta$ expresses the number of quantization levels of a $B$-bit quantizer within any interval $[a, b]$. Writing $\Pi_i$ for the $i$th quantization cell and $\Pi(\theta)$ for the width of the cell containing $\theta$, we can write the mean square error of quantized weights as

$$\mathbb{E}|\theta - \theta^q|^2 = \sum_{i=1}^{2^B} \mathbb{P}[\theta \in \Pi_i]\, \mathbb{E}[|\theta - \theta_i^q|^2 \mid \theta \in \Pi_i]$$

$$\overset{(a)}{\approx} \sum_{i=1}^{2^B} \mathbb{P}[\theta \in \Pi_i] \frac{|\Pi_i|^2}{12} \overset{(b)}{\approx} \int p_\theta(\theta) \frac{\Pi^2(\theta)}{12}\, d\theta \tag{14}$$

$$\overset{(c)}{\approx} \frac{1}{2^{2B}} \int p_\theta(\theta) \frac{\lambda^{-2}(\theta)}{12}\, d\theta$$

in which (a) follows from our assumption that weight distribution is approximately uniform within each quantization cell, (b) follows from an integral approximation of the finite sum, and (c) follows from the relationship $2^B \lambda^{-1}(\theta) = \Pi(\theta)$, all of which hold approximately when $B$ is sufficiently large.

To find the density $\lambda$ of quantization levels that leads to the minimum quantization error when $\theta$ has density $p_\theta$, we use Hölder's inequality: $\int p_\theta^{1/3} \le (\int p_\theta \lambda^{-2})^{1/3} (\int \lambda)^{2/3}$. Since $\int \lambda = 1$, we have that $\int p_\theta \lambda^{-2} \ge (\int p_\theta^{1/3})^3$, which sets a lower bound on the last term of (14). This lower bound and hence minimum quantization error is attained iff $p_\theta \lambda^{-2} \propto \lambda$. The optimal density for quantization levels is therefore given by

$$\lambda(\theta) \propto p_\theta^{1/3}(\theta) \iff \Pi^{-1}(\theta) \propto p_\theta^{1/3}(\theta). \tag{15}$$

Rather than optimize the density $\lambda$ to minimize the quantization error for a given $p_\theta$, we could equivalently transform the weights $\theta$ as $\theta^\sigma = \sigma(\theta)$ via a non-linear $\sigma$, so that uniform quantization applied to $\theta^\sigma \sim p_{\theta^\sigma}$ leads to the same minimum quantization error. The width $\Pi(\theta)$ of non-uniform quantization cells quantizing $\theta$ relates to the width of uniform quantization cells of the companded (transformed) weights $\theta^\sigma = \sigma(\theta)$ as

$$d\sigma(\theta) = \frac{d\theta}{\Pi(\theta)} \propto p_\theta^{1/3}(\theta)\, d\theta \implies \sigma'(\theta) \propto p_\theta^{1/3}(\theta), \tag{16}$$

in which the first proportionality follows from (15). We can find the optimal nonlinear transform $\sigma$ by integrating $p_\theta^{1/3}(\theta)$ and normalizing (for convenience) the range of the integral to $[0, 1]$:

$$\sigma(\theta) = \left( \int_{-\infty}^{\infty} p_\theta^{1/3}(t)\, dt \right)^{-1} \left( \int_{-\infty}^{\theta} p_\theta^{1/3}(t)\, dt \right) \tag{17}$$

(Gersho & Gray, 1991). Finally, we obtain (8) by substituting the expression for the density of a Laplace distribution (parameterized by mean $\mu$ and standard deviation $S$) into $p_\theta$ above. Transform $\sigma$ is asymptotically optimal as $B \to \infty$ in (14).

## D. Algorithm Parameters

To aid the reproducibility of the results in Table 1, we document the code we used for all algorithms (RTN, GPTQ, OWQ, and AWQ) along with the command line arguments.

**RTN.** We use the OWQ code from `https://github.com/xvyaward/owq/tree/03cfc99` in the provided `owq` conda environment. In the case of e.g. Llama-2-7b-hf quantized to 3 bits, we run `python main.py meta-llama/Llama-2-7b-hf c4 --wbits 3 --nearest`.

**GPTQ.** We use the OWQ code from `https://github.com/xvyaward/owq/tree/03cfc99` in the provided `owq` conda environment. In the case of e.g. Llama-2-7b-hf quantized to 3 bits, we run the provided command `python main.py meta-llama/Llama-2-7b-hf c4 --wbits 3`. For results based on the group size of 256, we run `python main.py meta-llama/Llama-2-7b-hf c4 --wbits 3 -groupsize 256`.

**OWQ.** We use the OWQ code from `https://github.com/xvyaward/owq/tree/03cfc99` in the provided `owq` conda environment. In the case of e.g. Llama-2-7b-hf quantized to 3.01 bits, we run the provided command `python main.py meta-llama/Llama-7b-hf c4 --wbits 3 --target_bit 3.01`.

**AWQ.** We use the AWQ code `https://github.com/mit-han-lab/llm-awq/tree/3665e1a` in the provided `awq` conda environment. In the case of e.g. Llama-2-7b-hf quantized to 3 bits, we run the provided command `python -m awq.entry -model_path meta-llama/Llama-7b-hf --w_bit 3 --q_group_size 128 --run_awq --tasks`

`wikitext.`

## E. Output Produced By Different Quantized Models

Table 8 lists output produced by different quantized Llama-2-70b models in response to questions taken from the GSM8K dataset. For each question, a prompt is created by prepending the question text with five other question and target pairs from the dataset (known as a 5-shot evaluation). This allows the model to establish a context for the required output and format. It is interesting to note that severe quantization errors (as in the case of RTN) manifest as non sequiturs and errors in logic rather than unintelligible output.

## F. Convex Weight Pruning (Hassibi & Stork, 1992)

To facilitate comparison between convex weight quantization (this work) and the convex weight pruning work of Hassibi & Stork (1992), we provide a derivation of Hassibi & Stork's Optimum Brain Surgeon (OBS) algorithm (presented slightly differently), together with our commentary for additional clarification.

For simplicity, let us rewrite model (4) as $f(\,\cdot\,, \mathbf{\Theta}_1, \mathbf{\Theta}_2, \ldots, \mathbf{\Theta}_N) = f(\,\cdot\,, \mathbf{\Theta})$, where $\mathbf{\Theta}$ is a vector of all model weights across different layers of the model. The objective of convex weight pruning is to set some number of elements of $\mathbf{\Theta}$ to zero while fine-tuning the remaining elements to minimize the difference between the output of the pruned model $f(\,\cdot\,, \mathbf{\Theta}^p)$ and the output of the unpruned model $f(\,\cdot\,, \mathbf{\Theta})$. Writing the pruned weights as $\mathbf{\Theta}^p = \mathbf{\Theta} + \mathbf{\Delta}^p$, where $\mathbf{\Delta}^p$ is a vector of updates to be made to weights $\mathbf{\Theta}$, it is apparent that $\Delta_i^p = -\theta_i$ if the $i$th weight is to be pruned, otherwise $\Delta_i^p$ should be chosen to maximally compensate for the effect of other pruned weights on the output. Suppose we have decided to prune the $p$th element of $\mathbf{\Theta}$. The updated set of weights $\mathbf{\Theta}^p$ can be found by solving

$$
\begin{aligned}
\text{minimize} \quad & d(\mathbf{\Delta}^p) = \mathbb{E}_{\mathbf{X}} \| f(\mathbf{X}, \mathbf{\Theta} + \mathbf{\Delta}^p) - f(\mathbf{X}) \|_2^2 \approx \mathbb{E}_{\mathbf{X}} \big[ \mathbf{\Delta}^{pT} (J^T J)(\mathbf{X}, \mathbf{\Theta}) \mathbf{\Delta}^p \big] \\
\text{subject to} \quad & r(\mathbf{\Delta}^p) = \mathbf{e}_p^T \mathbf{\Delta}^p - \theta_p = 0
\end{aligned}
\tag{18}
$$

in which $J(\mathbf{X}, \mathbf{\Theta})$ represents the Jacobian of $f(\mathbf{X}, \mathbf{\Theta})$ with respect to $\mathbf{\Theta}$, and $\mathbf{e}_p^T$ is an operator that picks out the $p$th element of a vector. The Lagrangian of this problem becomes

$$
\mathscr{L}(\mathbf{\Delta}^p, \lambda) = \frac{1}{2} \mathbb{E}_{\mathbf{X}} \big[ \mathbf{\Delta}^{pT} (J^T J)(\mathbf{X}, \mathbf{\Theta}) \mathbf{\Delta}^p \big] + \lambda (\mathbf{e}_p^T \mathbf{\Delta}^p - \theta_p)
\tag{19}
$$

in which $\lambda$ represents the dual variable associated with the equality constraint $\mathbf{e}_p^T \mathbf{\Delta}^p - \theta_p = 0$.

To solve (18), we differentiate $\mathscr{L}$ with respect to $\mathbf{\Delta}^p, \lambda$ and set all obtained derivatives equal to 0 to obtain the first-order optimality conditions $\mathbb{E}_{\mathbf{X}} \big[ (J^T J)(\mathbf{X}, \mathbf{\Theta}) \big] \mathbf{\Delta}^p + \mathbf{e}_p \lambda = \mathbf{0}$ and $\mathbf{e}_p^T \mathbf{\Delta}^p - \theta_p = 0$. After some algebraic manipulations, we obtain the optimizing values

$$
\mathbf{\Delta}^p = -\mathbb{E}_{\mathbf{X}} \big[ (J^T J)(\mathbf{X}, \mathbf{\Theta}) \big]^{-1} \mathbf{e}_p \lambda, \qquad \lambda = -\frac{\theta_p}{\mathbb{E}_{\mathbf{X}} \big[ (J^T J)(\mathbf{X}, \mathbf{\Theta}) \big]_{pp}^{-1}},
\tag{20}
$$

in which the expression for $\lambda$ is obtained by substituting the expression for $\mathbf{\Delta}^p$ above into the second optimality condition $\mathbf{e}_p^T \mathbf{\Delta}^p - \theta_p = 0$ and solving for $\lambda$. Combining both expressions finally produces an update $\mathbf{\Delta}^p$ that minimizes the objective in (18):

$$
\mathbf{\Delta}^p = -\frac{\theta_p}{\mathbb{E}_{\mathbf{X}} \big[ (J^T J)(\mathbf{X}, \mathbf{\Theta}) \big]_{pp}^{-1}} \mathbb{E}_{\mathbf{X}} \big[ (J^T J)(\mathbf{X}, \mathbf{\Theta}) \big]^{-1} \mathbf{e}_p, \qquad d(\mathbf{\Delta}^p) = \frac{1}{2} \frac{\theta_p^2}{\mathbb{E}_{\mathbf{X}} \big[ (J^T J)(\mathbf{X}, \mathbf{\Theta}) \big]_{pp}^{-1}}.
\tag{21}
$$

So far, we assumed that we were given the index $p$ of the weight to prune from $\mathbf{\Theta}$. To actually pick the best weights to prune away, we can compute the pruning loss $d(\mathbf{\Delta}^i)$ for all indices $i$, picking the index $i$ associated with minimum loss. That is,

$$
p = \underset{i}{\arg\min} \, \frac{1}{2} \frac{\theta_i^2}{\mathbb{E}_{\mathbf{X}} \big[ (J^T J)(\mathbf{X}, \mathbf{\Theta}) \big]_{ii}^{-1}},
\tag{22}
$$

after which $\mathbf{\Delta}^p$ can be calculated (and consequently $\mathbf{\Theta}^p$), and further weights to prune can be picked by initializing $\mathbf{\Theta} \leftarrow \mathbf{\Theta}^p$ and repeating the process until some pruning criterion has been met.

**Table 8: Output to questions from the GSM8K dataset (test) by 3-bit-quantized Llama-2 70B.** Scores computed across 1319 answers obtained using the flexible extract filter.

| Question | Target | FP16 (Score: 53.90%) | RTN (Score: 6.14%) | GPTQ (Score: 46.47%) | Ours (Score: 48.60%) |
|---|---|---|---|---|---|
| Question: Every day, Wendi feeds each of her chickens three cups of mixed chicken feed, containing seeds, mealworms and vegetables to help keep them healthy. She gives the chickens their feed in three separate meals. In the morning, she gives her flock of chickens 15 cups of feed. In the afternoon, she gives her chickens another 25 cups of feed. How many cups of feed does she need to give her chickens in the final meal of the day if the size of Wendi's flock is 20 chickens?\nAnswer: | If each chicken eats 3 cups of feed per day, then for 20 chickens they would need 3*20=<<3*20=60>>60 cups of feed per day.\nIf she feeds the flock 15 cups of feed in the morning, and 25 cups in the afternoon, then the final meal would require 60-15-25=<<60-15-25=20>>20 cups of chicken feed.\n#### 20 | Wendi feeds each chicken 3 cups of feed per day.\nShe feeds her flock 15 cups of feed in the morning and 25 cups of feed in the afternoon.\nThe total amount of feed she gives her flock is 15 cups + 25 cups = <<15+25=40>>40 cups.\nSince each chicken gets 3 cups of feed per day, the total number of chickens in her flock is 40 cups / 3 cups = <<40/3=13.33333333333333 33333333333333333333333 33333333333333333333333 33333333333333333333333 33333333333333333333333 3333333333333333333333 | 15 cups + 25 cups + 10 cups = <<15+25+10=50>>50 cups of feed.\n#### 50\n | 15 cups in the morning + 25 cups in the afternoon = <<15+25=40>>40 cups of feed\nEach chicken gets 3 cups of feed, so 40 cups of feed / 3 cups per chicken = <<40/3=13.33>>13.33 chickens\nSince there are 20 chickens, the number of cups of feed in the final meal is 20 - 13.33 = <<20-13.33=6.67>>6.67 cups\n#### 6.67\n\n | 15 cups + 25 cups = <<15+25=40>>40 cups of feed\n20 chickens x 3 cups = <<20*3=60>>60 cups of feed\n60 cups - 40 cups = <<60-40=20>>20 cups of feed\n#### 20\n\n |
| Question: A robe takes 2 bolts of blue fiber and half that much white fiber. How many bolts in total does it take?\nAnswer: | It takes 2/2=<<2/2=1>>1 bolt of white fiber\nSo the total amount of fabric is 2+1=<<2+1=3>>3 bolts of fabric\n#### 3 | 2 + 2/2 = <<2+2/2=3>>3 bolts\n#### 3\n\n | 2 blue bolts + 2 white bolts = <<2+2=4>>4 bolts in total.\n#### 4\np | 2 + 1 = <<2+1=3>>3 bolts\n#### 3\n\n | 2 + 2/2 = <<2+2/2=3>>3 bolts\n#### 3\n\n |
| Question: Josh decides to try flipping a house. He buys a house for $80,000 and then puts in $50,000 in repairs. This increased the value of the house by 150%. How much profit did he make?\nAnswer: | The cost of the house and repairs came out to 80,000+50,000=$<<80000+50000=130000>>130,000\nHe increased the value of the house by 80,000*1.5=<<80000*1.5=120000>>120,000\nSo the new value of the house is 120,000+80,000=$<<120000+80000=200000>>200,000\nSo he made a profit of 200,000-130,000=$<<200000-130000=70000>>70,000\n#### 70000 | The house is now worth 150% of $80,000, which is $80,000 * 150/100 = <<80000*150/100=120000>>$120,000.\nSo he made $120,000 - $80,000 - $50,000 = <<120000-80000-50000=20000>>$20,000.\n#### 20000\n\n | Josh bought the house for $80,000 and spent $50,000 on repairs. So he spent $80,000 + $50,000 = $<<80,000+50,000=130,000>>$130,000 on the house.\nSo Josh spent $130,000 on the house.\n#### $130,000\n | Josh bought the house for $80,000 and put in $50,000 in repairs.\nThe value of the house increased by 150%, so the new value is 1.5*$80,000+$50,000 = $<<1.5*80,000+50,000=180,000>>180,000.\nSo Josh made $180,000 - $80,000 - $50,000 = $<<180,000-80,000-50,000=50,000>>50,000 in profit.\n#### 50,000\n\n | The value of the house increased by 150%, so it's now 150% of $80,000, which is $120,000.\nJosh spent $80,000 + $50,000 = <<80000+50000=130000>>$130,000 on the house, so he made $120,000 - $130,000 = <<120000-130000=-10000>>-$10,000.\n#### -10000\n\n |

| | | | | | |
|---|---|---|---|---|---|
| Kylar went to the store to buy glasses for his new apartment. One glass costs $5, but every second glass costs only 60% of the price. Kylar wants to buy 16 glasses. How much does he need to pay for them?", "answer | The discount price of one glass is 60/100 * 5 = $<<60/100*5=3>>3.\nIf every second glass is cheaper, that means Kylar is going to buy 16 / 2 = <<16/2=8>>8 cheaper glasses.\nSo for the cheaper glasses, Kylar is going to pay 8 * 3 = $<<8*3=24>>24.\nAnd for the regular-priced glasses, Kylar will pay 8 * 5 = $<<8*5=40>>40.\nSo in total Kylar needs to pay 24 + 40 = $<<24+40=64>>64 for the glasses he wants to buy.\n#### 64 | The first glass costs $5.\nThe second glass costs $5 * 60% = $<<5*60%=3>>3.\nThe third glass costs $5.\nThe fourth glass costs $5 * 60% = $<<5*60%=3>>3.\nThe fifth glass costs $5.\nThe sixth glass costs $5 * 60% = $<<5*60%=3>>3.\nThe seventh glass costs $5.\nThe eighth glass costs $5 * 60% = $<<5*60%=3>>3.\nThe ninth glass costs $5.\nThe tenth glass costs $5 * 60% = $<<5*60%=3>>3.\nThe eleventh glass costs $5.\nThe twelfth glass costs $5 * 60% = $<<5*60%=3>>3.\nThe thirteenth glass costs $5.\nThe fourteenth glass costs $5 * 60% = $<<5*60%=3>>3.\nThe fifteenth glass costs $5.\nThe sixteenth glass costs $5 * 60% | The cost of one glass is $5, and the cost of the second glass is $5 * $<<5*2=10>>10>>10 is the cost of the third glass.\nThe cost of the fourth glass is $5 * $<<5*3=15>>15 is the cost of the fifth glass.\nThe cost of the sixth glass is $5 * $<<5*4=20>>20 is the cost of the seventh glass.\nThe cost of the eighth glass is $5 * $<<5*5=25>>25 is the cost of the ninth glass.\nThe cost of the tenth glass is $5 * $<<5*6=30>>30 is the cost of the eleventh glass.\nThe cost of the twelfth glass is $5 * $<<5*7=35>>35 is the cost of the thirteenth glass.\nThe cost of the fourteenth glass is $5 * $<<5*8=40>>40 is the cost of the fifteenth glass.\nThe cost of the sixteenth glass is $5 * $<<5*9=45>>4 | The first glass costs $5.\nThe second glass costs $5 * 60% = $<<5*60%=3>>3.\nThe third glass costs $5 * 60% * 60% = $<<5*60%*60%=1.8>>1.8.\nThe fourth glass costs $5 * 60% * 60% * 60% = $<<5*60%*60%*60%=1.08>>1.08.\nThe fifth glass costs $5 * 60% * 60% * 60% * 60% = $<<5*60%*60%*60%*60%=0.648>>0.648.\nThe sixth glass costs $5 * 60% * 60% * 60% * 60% * 60% = $<<5*60%*60%*60%*60%*60%=0.3888>>0.3888.\nThe seventh glass costs $5 * 6 | The first glass costs $5.\nThe second glass costs 60% of $5, which is $<<5*60%=3>>3.\nThe third glass costs 60% of $5, which is $<<5*60%=3>>3.\nThe fourth glass costs 60% of $5, which is $<<5*60%=3>>3.\nThe fifth glass costs 60% of $5, which is $<<5*60%=3>>3.\nThe sixth glass costs 60% of $5, which is $<<5*60%=3>>3.\nThe seventh glass costs 60% of $5, which is $<<5*60%=3>>3.\nThe eighth glass costs 60% of $5, which is $<<5*60%=3>>3.\nThe ninth glass costs 60% of $5, which is $<<5*60%=3>>3.\nThe tenth glass costs 60% of $5, which is $<<5*60%=3>>3.\nThe eleventh glass costs 60% |
| Question: Janet's ducks lay 16 eggs per day. She eats three for breakfast every morning and bakes muffins for her friends every day with four. She sells the remainder at the farmers' market daily for $2 per fresh duck egg. How much in dollars does she make every day at the farmers' market?\nAnswer: | Janet sells 16 - 3 - 4 = <<16-3-4=9>>9 duck eggs a day.\nShe makes 9 * 2 = $<<9*2=18>>18 every day at the farmer's market.\n#### 18 | Janet eats 3 eggs for breakfast and bakes 4 muffins, so she uses 3 + 4 = <<3+4=7>>7 eggs.\nShe sells the remainder, so she sells 16 - 7 = <<16-7=9>>9 eggs.\nShe sells each egg for $2, so she makes $2 * 9 = <<2*9=18>>18 dollars.\n#### 18\n\n | Janet sold 16 eggs per day. She sold three eggs for breakfast every morning, so she sold 16 - 3 = <<16-3=13>>13 eggs for breakfast.\nShe sold four eggs for her friends every day, so she sold 16 - 4 = <<16-4=12>>12 eggs for her friends.\nShe sold twelve eggs at the farmers' market daily, so she sold twelve eggs at the farmers' market daily, so she sold twelve eggs at the farmers' market daily, so she sold twelve eggs at the farmers' market daily, so she sold twelve eggs at | Janet eats 3 eggs and bakes 4, so she sells 16 - 3 - 4 = <<16-3-4=9>>9 eggs.\nShe sells 9 eggs for $2 each, so she makes $2 * 9 = <<2*9=18>>18 dollars.\n#### 18\n\n | Janet eats 3 eggs for breakfast and bakes 4 muffins, so she uses 3 + 4 = <<3+4=7>>7 eggs.\nShe sells the remainder at $2 per egg, so she makes $2 * (16 - 7) = <<2*(16-7)=2*9=18>>$18 per day.\n#### 18\n\n |

