# OpenReview forum: "Radio: Rate–Distortion Optimization for Large Language Model Compression"
_ICML.cc/2025/Conference — ICML 2025 poster_

### Official Review · Reviewer_i5t4 · 2025-02-27

**Overall Recommendation:** 2

**Summary:**

The authors propose to utilize rate-distortion theory to guide the allocation when quantizing LLM. Through extensive tests on benchmark datasets and pre-trained models, it is shown that the proposed approach can provide improvement on the state of the art when quantizing LLMs for 3 or 4 bits per parameter on average.

## update after rebuttal

Thanks for the response.

For item (2) above, I feel the insistence on using a less efficient algorithm is not helping. Cover and Thomas did not provide an explicit algorithm but simply stated that it can be viewed as a dual variable. The algorithm I outlined earlier is well known,  is straightforward to implement, and is essentially nln(n) computation to find the exact solution. The current algorithm is essentially a first-order method, and we know that a first-order method converges slowly and only to an approximate solution. The additional "max" is not really an issue using standard convex optimization derivation, which is another dual variable. I do not believe this issue has been resolved in the rebuttal.

For (3) above, the authors were distracted and forgot to reply to my original question. The authors omitted the second-order term, but this is not well-justified because at the convergent solution, the first-order term would be almost zero, and it is not clear why the second-order term is indeed small relatively to the first-order term.

Also, for (3), I believe the authors are saying their approach is essentially very similar to the existing "Hessian-aware" approach but with an optimized rate allocation. I am not sure if the two allocations would differ significantly if the latter was applied on the finer grain level, and a more detailed comparison would help to clarify this.

Overall, the rebuttal did not change my view, because 1) the gain from the existing methods is relatively small, and I'm not particularly enthused; 2) the technique used is not significantly novel; and 3) there are still a few issues not fully resolved yet, as discussed above. I would keep the score unchanged, but will also not oppose strongly if the AC decides to recommend acceptance.

**Claims And Evidence:**

The authors claim three contributions.
1. A rate-distortion theoretic framework for quantizing LLMs
2. A stochastic ascent algorithm for solving the optimization problem.
3. Experimental results that show the proposed approach can offer improvements over the state of the art.

While I have some reservations about the first two claims, the last claim is reasonably well-supported.

For the first claim, the theoretical development under the R-D theory requires several approximations, and it is unclear if the key factor that enables the performance is actually the R-D theory formulation, or a better estimation of the statistics, or the grouping strategy, or the non-linear scaling (companding).

For the second claim, the problem actually has a straightforward "closed-form" solution. Under the simplification that the authors made, this reduces to the classical rate distortion theory on Gaussian sources, which would give the so-called water-pouring solution (see Cover Thomas' textbook). Here the problem also has an upper limit on the rate allocation, but this is also quite simple to address. The water-pouring solution can be solved in several ways, but it is well known that the overall complexity is linear in the number of groups (i.e., n), without a need for iteration over V. Given the known "closed-form" solution, I would not view the second claim as important or well-supported.

**Essential References Not Discussed:**

I would like to see the performance comparison includes Kim 2024.

**Experimental Designs Or Analyses:**

The experiments seem well designed for the purpose of verifying the performance, similar to other existing works. I would ask the authors to provide 1) a comparison on computation times of the quantization optimization with other methods such as OWQ, GuIP, AWQ and 2) a discussion on any computational advantage during inference, because some existing methods seem to quantize in other factorized forms and show improved inference computation advantages.

**Methods And Evaluation Criteria:**

The authors provide evaluations on benchmarks and well-known pretrained LLMs, which are well-accepted in the literature.

I have a concern on the general approach of using a calibration dataset to learn the optimal quantization rate allocation. This approach requires access to a trustworthy high-quality calibration dataset. Firstly, it seems unreasonable to assume such a dataset is available, as it needs to follow the same distribution as the training set, but the training dataset is usually unknown (e.g., the training set for LLama2). Secondly, the effect of the calibration dataset size should be carefully studied: a small dataset will not capture the function accurately, but too large a dataset will induce significant computational loss. In the extreme case where the training set is the same as the calibration set, the training itself should be redesigned to take into account the quantization, instead of as a post-training procedure.

**Other Comments Or Suggestions:**

N/A

**Other Strengths And Weaknesses:**

The area of LLM model compression is quite crowded at this point, and previous works seem to have obtained most of the potential compression gain already. Though the current work seems to provide some minor additional improvements, I'm somewhat less enthusiastic on the incremental improvement, unless the authors can point out the major novelty in the methodology or potential other computational advantages. It also seems that many existing works have already recognized the importance of allocating different bit depths for different groups of parameters depending on their sensitivity, e.g., OWQ and SqueezeLLM, and the eventually allocation strategies seem very similar to the proposed approach. It would help considerably if the authors can discuss the similarity and difference in more depths.

**Questions For Authors:**

See comments above.

**Relation To Broader Scientific Literature:**

N/A

**Theoretical Claims:**

One of the concerns mentioned above is on the theoretical derivation of the rate distortion expression. The authors made several approximations, which I am not convinced. One approximation is to throw away the second order term and only keep the first order term. When the training has converged, the first order derivative will be almost zero, and for this reason, it appears questionable to throw away the higher order terms in favor of the first order approximation. Some existing work emphasizes such an issue (e.g., Kim 2024), and uses the Hessian information but throws away the first order term instead. Another approximation is in (13), where the author argues that the quantization error is zero mean. While I intuitively agree this could be the case, the approximation error should be carefully analyzed, under proper assumptions.

---

> ### Author Rebuttal · Authors · 2025-04-01
>
> Thank you, especially for initiating a discussion around water-filling.
> ____
> *1. For the first claim, …, unclear if the key factor that enables the performance is actually the R-D theory formulation, or a better estimation of the stats ...*
>
> The manuscript already provides the brakdown of performance gains in Tables 3 and 2b–c. In Table 3, we start with the RTN scheme, add in the mixed precision depths, and finally companding, to derive the proposed method. In Table 2b, we show that estimation the statistics (gradient variances) is not that very sensitive to the number of tokens per sequence used. Our grouping strategy is similar to GPTQ’s, and we fix the group size at 512. Table 2c shows that changing group size has little effect on the quantized model accuracy.
> ____
> *2. For the second claim, the problem actually has a straightforward "closed-form" solution .. the so-called water-pouring solution (see Cover Thomas' textbook) ... without a need for iteration over V. Given the known "closed-form" solution, I would not view the second claim as important or well-supported.*
>
> In fact, Algorithm 1 (lines 14–16) already implements the water-filling algorithm. Your description of water-filling as a "closed-form" solution is not fully accurate. While the optimal rates are given in closed form for a given water level and source variances, the appropriate water level (equivalently, dual variable) V still must be found numerically, in order to satisfy the total rate constraint. Thus, even under the water filing framework, solving for V remains a necessary iterative component.
> ____
> *3. I have a concern on the general approach of using a calibration dataset to learn the optimal quantization rate allocation. This approach requires access to a trustworthy high-quality calibration dataset ....*
>
> In post-training quantization, the original training dataset is typically not assumed to be available. Prior work (e.g., Kim et al., 2024) has shown that a small calibration set — even ~100 random sentences — is sufficient for calibrating model quantization.
> ____
> *4. One of the concerns mentioned above is on the theoretical derivation of the rate distortion expression. The authors made several approximations, which I am not convinced. One approximation is to throw away the second order term and only keep the first order term. When the training has converged ...*
>
> It appears you are misreading (5). We are not discarding second-order terms of the distortion function. Rather, we linearize the model function f f inside the squared norm, which is equivalent to approximating the Hessian of the distortion by the outer product of the Jacobian of f. This is a standard approach in optimization; see Appendix B and also (Nocedal & Wright, 2009, p. 250).
> ____
> *5. Another approximation is in (13), where the author argues that the quantization error is zero mean ...*
>
> Quantization errors are not zero mean in practice. In fact, our manuscript (p. 4, last paragraph) explicitly states this. To address this, we compute the mean quantization error (bias) and update the model’s bias vectors accordingly after each quantization step. This correction is implemented in Algorithm 1, line 18.
> ____
> *6. The experiments seem well designed for the purpose of verifying the performance ... provide 1) a comparison on computation times of the quantization optimization with other methods such as OWQ, GuIP, AWQ and 2) a discussion on any computational advantage during inference.*
>
> Please refer to our response to DSG1.1 and DSG1.7, which include quantization runtimes and inference-time speedups across several model sizes.
> ____
> *7. The area of LLM model compression is quite crowded at this point … the authors can point out the major novelty …*
>
> We agree. Methods such as SqueezeLLM and OWQ can allocate 16 bits to more sensitive weights and 3 bits to others — but they do so heuristically. This naturally raises the question: why only 3 and 16 bits? Why not assign bit depths proportionally to sensitivity (e.g., from 0 up to 16 bits)? And what is the correct sensitivity metric for rate allocation?
>
> These questions are precisely what our rate–distortion framework is designed to address. It provides a principled method for allocating bit depths and selecting sensitivity metrics (gradient variance), as well as a numerical procedure for optimization. Without this foundation, bit allocation becomes a combinatorial search problem, rather than a tractable convex optimization.
> ____
> *8. I would like to see the performance comparison includes Kim 2024.*
>
> Thank you. They are now included in Tables 1 and 5, as well as  Table 6 in our response to DSG1.1.

---

> > ### Comment · Reviewer_i5t4 · 2025-04-05
> >
> > Thanks to the authors for the rebuttal response, which helped to clarify some issues.
> >
> > 1. "breakdown of performance gains in Tables 3 and 2b–c". This ablation study is not completely clear to me. Particularly the stepsize optimization is introduced before the depth optimization. It seems the order should be reversed, or both orders of ablation should be considered.
> >
> > 2. "Your description of water-filling as a "closed-form" solution is not fully accurate. While the optimal rates are given in closed form for a given water level and source variances, the appropriate water level (equivalently, dual variable) V still must be found numerically, in order to satisfy the total rate constraint. " This seems to be due to a misunderstanding on how water-filling solution can be computed by the authors. Since the R-D water-filling solution is a piece-wise defined function, only the (R,D) value (and corresponding water level) at the water-level thresholds need to be computed to determine which segment the target rate actually falls into. Then the explicit formula can be used for that segment. Based on this understanding, a simple algorithm can be designed without computing all the threshold values by combining with a bisection approach.
> >
> > 4. "It appears you are misreading (5). " I am referring to (12) in the appendix where it was stated "the second term of is approximately zero". In fact, if the authors are indeed approximating Hessian, what is the difference from the previous approach based on "Hessian-aware"?
> >
> > 8. Can you include the additional results in Table 1 and 5 and discuss whether your method provides any improvement?

---

> > > ### Author Response · Authors · 2025-04-08
> > >
> > > *1. breakdown of performance gains in Table 3 ... not clear to me.*
> > >
> > > The RTN scheme with MMSE step sizes is a well-established technique (see Lee et al., 2024, p. 13358, last para) and serves as the quantization backbone for Radio after it allocates bit depths. For this reason, we present RTN + MMSE immediately after MinMax RTN in Table 3—to isolate the improvement from MMSE step size optimization within the RTN framework, before introducing the additional gains from our bit-depth allocation strategy. Since Radio uses the RTN + MMSE step sizes as the backbone, this order of ablation more accurately reflects the logical progression of our method and enables a clearer decomposition of individual contributions.
> > > _____
> > > *2. … misunderstanding on how water-filling solution can be computed  ...*
> > >
> > > The water-filling procedure in Algorithm 1 (lines 11–14) follows the standard formulation in Cover & Thomas (2nd ed., Sec 10.3.2) and Boyd & Vandenberghe (Sec 5.1.6). These references treat water-filling as a convex optimization problem and solve for the optimal dual variable numerically. This approach is numerically robust, easy to implement, and consistent with standard treatment in the optimization literature.
> > >
> > > We do acknowledge that one could alternatively exploit the piecewise structure of the rate-water level function, by computing water-level thresholds and selecting the corresponding segment. However, under an additional maximum rate constraint, the water level generally cannot be computed in closed form and still requires iterative evaluation. As such, the potential advantage of the piecewise approach becomes more limited in our setting.
> > > _____
> > > *3. I am referring to (12) ... what is the difference from the previous approach based on "Hessian-aware"?*
> > >
> > > Thank you for clarifying. Yes, the second term of the Hessian in (12) is approximated to zero—this is a standard Gauss–Newton approximation in nonlinear least squares problems (Nocedal & Wright, eq 10.24–10.25).
> > >
> > > While our method, like HAWQ, leverages second-order statistics to guide quantization, the approaches differ fundamentally. HAWQ employs a custom notion of sensitivity (e.g., the largest eigenvalue of a block Hessian) to greedily assign bit depths at the layer level. However, such heuristics do not scale well to fine-grained quantization (e.g., per-channel or sub-channel needed for LLM compression), due to the combinatorial explosion of possible bit combinations to explore.
> > >
> > > By contrast, our approach is grounded in a rate–distortion framework that formulates bit allocation as a convex optimization problem. This enables bit-depth assignment at any level of granularity—including per-channel—in a tractable and principled manner. The manuscript already discusses this distinction in the last paragraph of Section 2 (just before Section 3, p. 2). In this sense, our method generalizes prior heuristics and offers both scalability and theoretical grounding.
> > > _____
> > > *4. Can you include the additional OmniQuant (Shao et al., 2024) and SqueezeLLM (Kim et al., 2024) results and discuss ...*
> > >
> > > Sure. Please see below for the results (due to the character limit, we only show 3-bit results; 4-bit results follow a similar trend). Note that calibration examples are sourced from the C4 dataset, so perplexity on C4 reflects how well a quantized model fits the calibration distribution. In contrast, WikiText2 perplexity reflects the model’s generalization ability to a different generation task and data distribution.
> > >
> > > Across all model sizes and families, Radio consistently outperforms both OmniQuant and SqueezeLLM on WikiText2, indicating stronger generalization. On C4, while performance is competitive across most settings, some perplexity degradation is observed with Meta LLaMA 2 models, which we attribute to Radio prioritizing generalization over potential overfitting to the calibration set (e.g., with the k-means quantization in SqueezeLLM).
> > >
> > > | Wikitext2 PPL | 125M | 350M | 1.3B | 2.7B | 6.7B | 13B | 30B | 66B | L2-7B | L2-13B | L2-70B |
> > > |-|-|-|-|-|-|-|-|-|-|-|-|
> > > 3-bit RTN | 1.3e3 | 64.57 | 119.47 | 298.00 | 23.54 | 46.04 | 18.80 | 6.1e3 | 6.66 | 5.52 | 3.98 |
> > > OmniQuant/128 | 32.25 | – | 15.71 | 13.18 |  11.27 | 10.47 | 9.79 | 9.53 | 6.03 | 5.28 | 3.78 |
> > > SqueezeLLM | – | – | 16.30 | 13.85 | 11.70 | 11.76 | 10.17 | – | 6.18 | 5.36 | 3.77 |
> > > Radio | **30.71** | **25.94** | **14.83** | **12.42** | **11.07** | **10.28** | **9.56** | **9.24** | 6.04 | **5.25** | **3.72** |
> > >
> > > | C4 PPL | 125M | 350M | 1.3B | 2.7B | 6.7B |	13B | 30B | 66B | L2-7B | L2-13B | L2-70B |
> > > |-|-|-|-|-|-|-|-|-|-|-|-|
> > > 3-bit RTN | 839.97 | 55.96 | 4.2e3 | 1.1e4 | 4.4e3 | 3.2e3 | 1.1e3 | 3.5e3 | 521.22 | 14.01 | 11.06 |
> > > OmniQuant/128 | 31.30 | – | 17.46 | 15.33 | 13.28 | 12.50 | 11.73 | 11.22 | 7.75 | 6.98 | 5.85 |
> > > SqueezeLLM | – | – | 17.19 | 15.62 | 13.41 | 13.55 | 11.85 | – | 7.72 | 6.97 | 5.73 |
> > > Radio | **30.05** | **26.20** | **16.88** | **14.91** | **13.14** | **12.35** | **11.62** | **11.19** | 8.04 | 7.22 | 5.99 |

---

### Official Review · Reviewer_DSG1 · 2025-03-13

**Overall Recommendation:** 4

**Summary:**

This paper proposed a rate-distortion optimization framework, Radio, for compressing large language models (LLMs) via quantization. Specifically, this paper formulate quantization as distortion problem and try to minimize it by assigning bit depth to weight groups.

**Claims And Evidence:**

Most claims are supported by empirical results across diverse models and tasks.

**Essential References Not Discussed:**

Please include more papers: SmoothQuant, OmniQuant, QuaRoT, ZeroQuant.

**Experimental Designs Or Analyses:**

This paper employs two key techniques:

- PCA
- Dual optimization

However, I wonder whether the time cost increases exponentially as the model size grows.

Additionally, for the proxy, can you explain why you choose gradient deviation? There are lots of proxies like Hessian, SNIP etc.

Can you show me the superiority of gradient deviation?

About the granularity, instead of per row or per column, here I use per channel or per token instead.

- Generally, most of method apply per channel quantization as per-channel is memory efficient which employ the locality of memory, while per-token would access memory across physical location.

For quantization, please add more experiments on SOTA LLMs like Qwen2.5. As there is evidence shows that the models under sufficient training will show distinct characteristics. (You can refer to paper: arxiv:2411.04330)

**Methods And Evaluation Criteria:**

The evaluation metrics (perplexity, QA accuracy) are appropriate

**Other Comments Or Suggestions:**

This paper did not follow the ICML paper instructions. Maybe the format is out of dated as there is no number of lines in this paper.

**Other Strengths And Weaknesses:**

The rate-distortion framework is a creative application of classical theory to modern LLM challenges.


Limited validation of theoretical assumptions; omission of deployment metrics (latency, hardware); insufficient discussion of calibration data sensitivity.

**Questions For Authors:**

- **Q1**: How does Radio’s quantization time scale with model size (e.g., OPT-175B), and how does it compare to GPTQ’s runtime?
- **Q2**: The derivation assumes uncorrelated quantization errors. How does this impact performance at ultra-low bit depths (e.g., 2 bits), and are there plans to address correlated errors?
- **Q3**: Why are hardware efficiency metrics (e.g., latency, memory bandwidth) absent? Including these would strengthen the case for deployment on resource-limited devices.

**Relation To Broader Scientific Literature:**

Radio builds on rate-distortion theory from signal compression and connects to prior LLM quantization works (GPTQ, AWQ). It addresses a gap by formalizing quantization as a resource allocation problem, contrasting with Hessian-based methods like OBS.

**Theoretical Claims:**

The derivations in Appendices B and C assume high bit depths and Gaussian/Laplace weight distributions.

---

> ### Author Rebuttal · Authors · 2025-04-01
>
> Thank you, especially for checking the derivation of (5)!
> ____
> *1. However, I wonder whether the time cost increases exponentially as the model size grows. How does Radio’s quantization time scale with model size (e.g., OPT-175B), and how does it compare to GPTQ’s runtime?*
>
> Thank you for raising this. The revised manuscript now includes Table 6 below, which shows that our runtime scales roughly linearly with model size — consistent with the complexity of other methods reported in the same table.
>
> **Table 6: Quantization running times.** We quantize the Meta Llama 2 family of LLMs to ~3 bits per weight on average and measure the running time of the proposed method. We also include the running times of GPTQ, QuIP, OWQ, AWQ, and SqueezeLLM.
>
> | Method       | Llama2-7B | Llama2-13B | Llama2-70B |
> |--------------|---------------|---------------| -------------- |
> | GPTQ/256 (Frantar et al., 2022)  | 10m    | 18m | 90m |
> | QuIP (Chee et al., 2024) | 36m    | 80m | 9h |
> | OmniQuant/128 (Shao et al., 2024) | 66m    | 132m | 15h |
> | SquezeLLM (Kim et al., 2024) | 11m    | 23m | 92m |
> | Radio (Ours) | 47m    | 97m | 11h |
> -----------------------------------------------
> ____
> *2. Additionally, for the proxy, can you explain why you choose gradient variance? There are lots of proxies like Hessian, SNIP etc. Can you show me the superiority of gradient deviation?*
>
> This is a great question. In (5) and Appendix B, we show that gradient variance arises naturally from the rate–distortion optimization formulation. Under this framework, it is the theoretically correct sensitivity measure to incorporate. In contrast, heuristics like SNIP are not grounded in the rate–distortion theory that underpins our formulation.
> ____
> *3. About the granularity, most of method apply per channel quantization as per-channel is memory efficient which employ the locality of memory, while per-token would access memory across physical location.*
>
> Agreed. Our method already uses per-channel quantization, not per-token quantization. Moreover, our current focus is on weight quantization, where per-token quantization (relevant for activations) does not apply.
> ____
> *4. For quantization, please add more experiments on SOTA LLMs like Qwen2.5 ...*
>
> While Qwen and DeepSeek may soon overtake Llama 2/3 in adoption, they were unpublished at the time of our ICML submission, and even now, most SOTA quantization methods (e.g., OWQ, QuIP, SqueezeLLM) do not yet support them. For reproducibility and fair comparison, we based our experiments on OPT and Llama 2 (11 models in total), which are widely used baselines in recent quantization works.
> ____
> *5. Please include more papers: SmoothQuant (Xiao et al., 2023), OmniQuant (Shao et al., 2024), QuaRoT (Ashkboos et al., 2024), ZeroQuant (Yao et al.., 2023).*
>
> Thank you for the suggestions. The revised manuscript now discusses all these works. Specifically, Section 2, paragraph 3 now reads:
>
> “...by re-scaling or retaining original weight values (Lin et al., 2024; Xiao et al., 2023), low-rank decomposition of quantization error matrices (Shao et al., 2024), and decorrelation of weight matrices prior to quantization (Ashkboos et al., 2024).”
>
> For OmniQuant, we also include its 3-bit and 4-bit results on OPT and Llama 2 models in Tables 1 and 5.
> ____
> *6. The derivation assumes uncorrelated quantization errors. How does this impact performance at ultra-low bit depths (e.g., 2 bits), and are there plans to address correlated errors?*
>
> Indeed, at very low bit depths, correlation in weight quantization errors comes more pronounced. One way to decorrelate quantization errors is by decorrelating the weights themselves prior to rate quantization and quantization (Gersho and Gray, ). We now include this discussion in Sec 5 para 2: "It is known that scalar quantization of correlated sources can be improved by decorrelating the sources prior to quantization (see Gersho and Gray, 1991). This will be explored in future work.”
> ____
> *7. Why are hardware efficiency metrics (e.g., latency, memory bandwidth) absent? ...*
>
> The revised manuscript now includes Table 7 shown below, detailing the acceleration achieved by 3-bit-average weight quantization for different models / embedding dimensionality. Sec 4 (last para) now reads: “Timing results. ... Table 7 lists the acceleration achieved by our custom quantized matrix-vector multiply kernel, with acceleration ranging between 1.4 and 3.3 depending on the embedding dimensionality.
>
> **Table 7: Acceleration  due to quantized mat-vector multiplies relative to multiplication in FP16.** A 3-bit weight matrix of dimension N×M multiplies a vector of length M to produce a vector of length N (denoted M→N).
>
> | Model (Embedding)       | E→ E| E→ 4E | 4E→ E | Overall |
> |--|--|--|--|--|
> | OPT-1.3B (E=1024)| 0.9|2.1|2.7|1.4|
> | OPT-6.7B (E=4096)| 2.4|3.1|3.1|2.8|
> | OPT-30B (E=7168)| 3.2 |3.2|3.1|3.2|
> | OPT-175B (E=12288)|3.2|3.2|3.8|3.3|
> ------------------------------------------

---

> > ### Comment · Reviewer_DSG1 · 2025-04-03
> >
> > Thanks the authors for their detailed rebuttal. Most of my concerns are well addressed. I have raised my score.

---

### Official Review · Reviewer_dfYh · 2025-03-13

**Overall Recommendation:** 3

**Summary:**

The paper
* formulates a rate-distortion theoretic framework for quantization of LLMs
* designs an algorithm to solve the optimization (for model compression) resulting from that framework
* runs the model compression method on various models to show the aspects

## update after rebuttal
My assessment won't change. Overall the paper looks OK to me and the author does elaborate on the point I pick.

**Claims And Evidence:**

Yes

**Essential References Not Discussed:**

Not as I'm aware of.

**Experimental Designs Or Analyses:**

* `WikiText2 perplexity` looks fair since the only moving part is the compression method while the authors' codebases are used for all the other methods to compare
* `Effect of hyperparameters on quantized model accuracy` shows the trend
* `Perplexity across optimization iterations` (Figure 4) is not very well explained
* `Ablations and pruning effects of quantization` looks good
* `2.x-bit quantization and downstream tasks` `C4 perplexity (validation)` look fair

**Methods And Evaluation Criteria:**

Yes

**Other Comments Or Suggestions:**

0

**Other Strengths And Weaknesses:**

The rate-distortion framework borrows idea from classic information theory which makes the method principled.

**Questions For Authors:**

0

**Relation To Broader Scientific Literature:**

* bit depth determination has been a topic in LLM quantization

**Theoretical Claims:**

The paper
* defines the bit depth optimization problem in eq.3
* makes a relaxation and derives the first-order condition eq.4
* gets an approximation of the partial derivative in eq.5 then combine that with eq.4 to get the update rule eq.6
* formula for stochastic is in eq.7
* eq.8 shows an an asymptotically optimal choice of sigmoid function under the assumption of weights being Laplace-distributed
* eq.9 gives the theoretical gain from weight grouping with the claim of being non-negative

The math looks good.

---

> ### Author Rebuttal · Authors · 2025-04-01
>
> Thank you, especially for checking through the individual equations.
> ____
> *1. Perplexity across optimization iterations (Figure 4) is not very well explained.*
>
> Thank you for pointing this out. Figure 4 shows how quantized model accuracy (in terms of perplexity) improves as more gradient variances are accumulated and weights are re-quantized accordingly. We observe that performance saturates after about 30 accumulation iterations, beyond which additional iterations yield diminishing returns. This is valuable, as accumulating gradient variances via backpropagation can be computationally expensive — thus, the ability to early-stop after ~30 iterations is a practical benefit.

---

### Official Review · Reviewer_EoEV · 2025-03-19

[review text omitted: it was posted to a different submission]

---

> ### Author Rebuttal · Authors · 2025-04-01
>
> Thank you! First, we kindly clarify that our work is entitled “Radio”, not “CMPQ”. CMPQ appears to refer to a prior work by different authors.
> ____
>
> *1. The introduction of multiple components such as channel-wise quantization, non-uniform quantization, and two types of outlier protection may make the method relatively complex to implement and understand...*
>
> We respectfully disagree. As shown in Algorithm 1 and detailed in Appendix A, our method is straightforward to implement. Notably, our non-uniform quantization is achieved via a simple companding followed by uniform quantization — a lightweight alternative to more complex approaches like Lloyd-Max quantization. Furthermore, Radio does not involve any explicit outlier protection mechanism; perhaps this was conflated with another work.
> ____
> *2. Although the paper compares [Radio] with several relevant baselines, there may be other emerging quantization techniques that could be included in the comparison to further validate the superiority of [Radio].*
>
> Thank you! As also requested by other reviewers, we add the quantization results of SqueezeLLM (Kim et al., 2024) and OmniQuant (Shao et al., 2024) in Tables 1, 5 as well as their timing results. Please see our response to DSG1.5 and i5t4.8.
> ____
> *3. The authors could further explore the generalization of [Radio] to other types of language models ...*
>
> Thank you! Low-bit quantization of Qwen and DeepSeek is currently planned. Their quantized model accuracy and performance will be shared directly on our project webpage (url to be revealed after double-blind review). Please also see our response to DSG1.4
> ____
> *4. Have you considered combining [Radio] with other model compression techniques, such as pruning, to further improve the efficiency of LLMs?*
>
> Yes — interestingly, Radio implicitly induces mild pruning through zero-bit quantization, as shown in Table 3b. Additionally, our method can be combined with weight matrix decorrelation before bit allocation and quantization. This is now discussed in Sec. 5, para. 2, where we mention that decorrelating correlated sources can improve scalar quantization (cf. Gersho and Gray, 2991). We plan to explore rate–distortion optimal quantization of decorrelated weights in future work. Please also see our response to DSG1.6
> ____
> "5. Could you provide more insights into the selection of the specific percentages for outlier protection and how sensitive the performance of [Radio] is to these values?"
>
> [Radio] does not use any explicit thresholding or masking for outlier protection. Instead, our companding function (Eq. 8 and Appendix C) continuously adjusts the quantization resolution across the distribution. This adaptively mitigates quantization error at the center while preserving dynamic range in the tails (outliers) — see Fig. 2 for a visualization.

---

### Decision · Program_Chairs · 2025-05-01

**Decision:**

Accept (poster)

**Comment:**

This paper studies model compression with a technique that is inspired by the information-theoretic viewpoint of rate distortion theory. The authors introduce a new weight compression approach based on dual ascent. There are also several additional tricks used, e.g., choosing certain clusters of weights for the quantization granularity. This leads to reasonably strong results (improved perplexities for 3/4-bit quantization of LLMs versus a variety of compression baselines).

The reviewers are broadly positive; there are enough interesting new elements in this paper that, along with good performance, makes it worth including. The main questions that came up are around complexity/time for the technique, which the authors have addressed. Additionally, as noted by reviewer i5t4, the authors should do a bit more clarification on what is really new here compared to some of the standard algorithms like water-filling. Nevertheless, there is plenty here that is interesting and worth accepting.